



# Dynamic robust active wake control

Stoyan Kanev[1] and Edwin Bot[1]

[1]TNO Energy Transition, Wind Energy, Westerduinweg 3, 1755LE Petten, Netherlands

**Correspondence:** Stoyan Kanev (stoyan.kanev@tno.nl)

**Abstract.** Active Wake Control (AWC) is a strategy for operating wind farms in a way to maximize the overall power production and/or reduce structural loading on the wind turbines. Many recent studies indicate that this technology, and more specifically the so-called wake redirection approach to AWC, have a significant potential for increasing the annual energy production (AEP) by up to a few percentage points. The current state-of-the-art approach is to optimize AWC for a range of
static wind conditions, which is expected to perform sub-optimally in real-life due to the continuous variations of the wind resource and the very slow yaw dynamics of the turbines. Recent work has addressed this variability in a robust design setting with the focus on maximizing the energy capture (robust AWC). This paper continues on this line of research, and develops a *dynamic* robust AWC strategy that aims to optimize the balance between maximum power production (requiring increased level of yawing) and minimum loads on the yaw drive (requiring limited yaw motion). It is shown with a realistic case study
that the developed dynamic robust AWC can result in a large reduction of the loading on the yaw drive while at the same time improving the overall power gain, as compared to the conventional nominal AWC.

## 1   Introduction

During the last decade, the field of active wake control (AWC) has been widely studied by many researchers worldwide. AWC is an approach to operate the wind turbines in a wind farm in a collaborative manner with the aim of reducing the negative
effects of the wakes behind wind turbines on the overall power production and the individual wind turbines' structural loading (Andersson et al., 2021; Kheirabadi and Nagamune, 2019; Boersma et al., 2017). More specifically, the wake redirection approach to AWC, employing intentional yaw misalignment to steer wakes away from downstream turbines, is currently considered as the most potential technology with respect to power production increase, with possible power gains of up to a few percent on annual basis (Kanev et al., 2018; Gebraad et al., 2017; Fleming et al., 2016).
At present, large scale implementation of wake redirection AWC is hampered by two main challenges. The first one concerns the increased risk perception due to the required operation under significant yaw misalignment, driven by the lack of comprehensive understanding of the impact on the structural loading on the turbines, often in combination with restrictive obligations in service contracts. The second challenge is related to the uncertainty in the predictions for the expected annual energy production (AEP) increase, caused by the simplistic static approach that is currently used to optimize AWC on the one
side, and the underlying uncertainties in the modelling used for that purpose on the other side. Within the past few years, some initial results addressing these challenges have started to appear in the literature. With respect to the first one, for instance, most





existing studies are limited to just one or two turbines (Fleming et al., 2013, 2015; Croce et al., 2020; Zalkind and Pao, 2016). A more detailed study involving a utility scale wind farm was presented in Kanev et al. (2020), where the impact of wake steering AWC on the structural loads of the turbines during their complete lifetime has been investigated using the so-called
loads lookup table (LUT) approach (Reyes et al., 2020). The results demonstrate that, even though by itself yaw misalignment does increase the structural loads of some turbines in specific wind conditions, the wake-induced loading is decreased even more, so that the accumulated loads over the whole lifetime of each wind turbine generally remain lower than without AWC. This conclusion is implicitly confirmed by the fact that the industry starts to develop this technology into commercial products (Siemens Gamesa Renewable Energy, 2019).

To sort out the second challenge, however, further research is required. The uncertainty in the AEP predictions is driven by a wide range of factors: the variability in the wind resource, wake meandering, uncertainties in the environmental conditions (atmospheric stability, turbulence, shear, veer, etc.), the slow dynamics of the yaw system, wake model uncertainties, measurement uncertainties, etc. Disregarding these, as done in the current state-of-the-art approach to AWC design that relies on optimization under stationary conditions, gives rise to suboptimal performance in real-life, i.e. lower power gain or, possibly
even loss of power. To ensure maximum power gain in the field, it is essential to move from the current static approach to dynamic robust AWC design that properly accounts for the mentioned variabilities and uncertainties. One way to achieve that is by means of a two-stage approach, where the term "robust" relates to the selection of static yaw misalignment set-points that maximize power production in the presence of uncertainties (either deterministic, or stochastic), while the "dynamic" part deals with the adaptation of the set-points based on the highly varying wind resource measurements aiming to optimize the
balance between maximal power gain and minimal loading on the yaw drive. The importance of achieving both robustness and optimized yaw dynamics has become apparent in recent field studies with wake redirection (Fleming et al., 2020, 2019).

The topic of robust AWC design has already attracted the attention of some researchers. Quick et al. (2017) considered yaw angle uncertainty into the optimization for the yaw misalignment set-points. The uncertainty was modelled in a stochastic setting assuming Gaussian probability density function (PDF). This work was followed by Rott et al. (2018), where robustness
with respect to wind direction variability and wind direction measurement uncertainty was considered, both modelled through a single PDF. Later on, in 2020, two relevant publications appeared. In the first one, Simley et al. (2020) presented results on robust AWC optimization including uncertainty in both the wind direction and the yaw position. They evaluated the performance of the robust AWC in case studies with different wind turbine spacings and turbulence intensities. The second work was published by Quick et al. (2020), and considers robustness with respect to a range of uncertain parameters, namely wind speed,
wind directions, turbulence intensity, wind shear and yaw angle. The authors used a polynomial chaos expansion approach to deal with the underlying high computational complexity of the resulting optimization, and demonstrated that the wind direction is the most significant contributor to uncertainty in the power predictions. Above-mentioned studies on robust AWC were all performed using a simplified control-oriented wake model, namely the FLOw Redirection and Induction in Steady State (FLORIS) model – an understandable choice given the computational requirements for robust optimization. Finally, Howland
(2021) studied AWC robustness with respect to model parameter uncertainty and wind direction uncertainty, utilizing a different steady-state wake model, called the lifting line model. The author uses an Ensemble Kalman filter to estimate the relevant



wake model parameters and the corresponding probability distributions. In the simulation setup considered there it appeared that including model uncertainty gives rise to a more significant improvement than wind direction uncertainty. It should also be pointed out that, as an alternative to the robust design of yaw offset set-points that are valid for a range of conditions, one could

consider adaptive solutions in which the yaw set-points are updated in real-life based on (estimates of) the actual operating conditions. Such an approach is pursued in Howland et al. (2020), where a gradient ascent algorithm updates the yaw offsets at each iteration based on analytically derived gradients for the lifting line wake model, the parameters of which are estimated online.

With respect to dynamic AWC, there is only very limited research published so far. There are, however, numerous publi-

cations in which dynamic (or quasi-static) simulations are performed with AWC, that include the yaw system dynamics and variations is the wind conditions. However, the parameters of the AWC adaptation algorithm in these studies, typically a low pass filter acting on the wind direction and wind speed, are not optimized with respect to the power gain and/or the yaw duty. In Simley et al. (2020), for instance, dynamic simulations have been performed using the stationary FLORIS wake model by propagating the low-frequency components of the wind through the model (neglecting spatiotemporal inflow variations), and

adding high-frequency components on top of those at turbine level to feed the yaw system model. In terms of dynamics, a more realistic wake model is used by Bossanyi (2018) that includes turbine and wake dynamic effects and utilizes a stochastic wind field correlated across the wind farm. The dynamic model has been used to simulate AWC and evaluate the impact on farm power and turbine loads. However, even though the authors state to have done a few iterations in choosing parameters of the dynamic AWC algorithm that appear to work reasonably, the presented study does not extend to the point of optimizing

the AWC parameters with respect to energy production and yaw duty. In a different work, the same author demonstrates that a centralized yaw control strategy, in which information from surrounding wind turbines is used in the yaw control algorithm, can lead to a drastic reduction in the yaw duty and increase the power capture at the same time (Bossanyi, 2019). Even though AWC is not considered in that study, it is mentioned by the author that the proposed yaw control strategy would be beneficial in AWC as well. Finally, in Kanev (2020) some initial results with optimizing the parameters of a dynamic AWC have been

presented. Although the findings there clearly support the necessity of properly optimizing the dynamics of the AWC algorithm, the conclusions there remain of limited value due to the simplified modelling approach employed. More specifically, the stationary FLORIS model was used there, extended with a simple time delay model representing wake dynamics. As such, this model is quite unrealistic as it completely neglects spacial inflow variations. Also missing in the modelling approach there is the impact of the increased turbulence in front of waked turbines on their wind measurements, and the resulting increased

turbine yawing at downstream turbines.

The present work extends on above mentioned research in focusing on dynamic robust AWC. The first part of the study concerns *robust* design and analysis using stationary simulations with FarmFlow, a 3D parabolized Reynolds-averaged Navier Stokes code with prescribed pressure gradients and $k - \epsilon$ turbulence model (Bot and Kanev, 2020). The starting point is the selection of varying or uncertain quantities that can affect the performance of the AWC. To this end, an uncertainty quantifi-

cation analysis has been performed for a range of variables (wind speed, wind direction, yaw error, turbulence intensity, wind shear, air density, power curve, thrust curve, power loss coefficient due to yawed error). This analysis indicated that the highest



uncertainty contributors are the wind direction, yaw error, turbulence intensity and the wind velocity. Subsequently, robust yaw misalignment set-points have been optimized with respect to uncertainties in these parameters, modelled as independent stochastic processes with selected PDFs. A stationary analysis based on stochastic averaging indicated that the robust AWC
design slightly outperforms the nominal one in terms of power gain.

With the robust AWC in place, the second part of this study continues with the *dynamic* design and analysis. To this end, the originally stationary FarmFlow wake model has been extended to enable dynamic simulations, including wake dynamics and a dynamic yaw control model. The dynamic simulation model is fed by a realistic wind field including temporal and spacial inflow variations, that include both micro-scale (fast turbulence variations ranging up to several hundreds of meters) and meso-
scale (slow variations extending to ten kilometres and more), with corresponding coherence functions and cross power spectra that relate the stochastic properties between different points in space, and including terms to model the flow advection. The yaw dynamics are modelled at a faster sample rate than the wake model, and an additional higher frequency stochastic signal is added to the yaw error to model the increased noise in the yaw error measurements that enter into the yaw position controller. The size of this added noise is made dependent on the turbulence intensity in the flow in front of the turbine. This gives rise to
an increased yaw activity of downstream wind turbines, as seen in real-life measurements. Next, an AWC dynamic adaptation algorithm is considered, consisting of a low-pass (LP) filter, a hysteresis, and sample and hold mechanism, similar to (Kanev, 2020). Numerous dynamic simulations have been performed with different dynamic adaptation parameters, both with nominal and robust AWC yaw set-points. Based on the results from these simulations, the optimal parametrization of the dynamic AWC are determined. The resulting dynamic robust AWC is shown to deliver a large reduction in the yaw duty in combination with
increase in the power gain as compared to the nominal AWC solution.

The remaining part of the manuscript is organized as follows. The next section summarizes the modelling used in the study. Sect. 3 outlines the the uncertainty quantification analysis, the selection of the dominant uncertainties, the design of robust AWC with respect to these and, finally, gives the results from a stationary robust analysis. Next, the optimization of the AWC dynamic adaptation algorithm is discussed in Sect. 4. Sect. 5 goes on with demonstrating the benefits from the developed dynamic robust
AWC methodology on a case study with a model of an existing offshore wind farm. The manuscript is concluded with some final remarks in Sect. 6.

## 2 Wind farm model

The wind farm simulation model consists of a stochastic wind field generator, wake model (dynamic FarmFlow), model of the wind turbines' yaw systems, and dynamic robust AWC. A block scheme of the simulation model is given in Fig. 1. The different
line colors are meant to indicate different sample times. The base sampling rate (black lines) is set by the wind field time series (typically $0.1Hz$ or slower). The yaw model operates at faster sampling rates (green lines) to enable realistic modelling of the yaw motion (e.g. $1Hz$), which is important to assess the impact of AWC on the yaw duty. The actual simulation model has a much more extensive interface between the wake model and the AWC algorithm enabling a wide range of possible future applications, but is visualized here in a simplified way, sufficient for the present discussion. Finally, part of the AWC algorithm



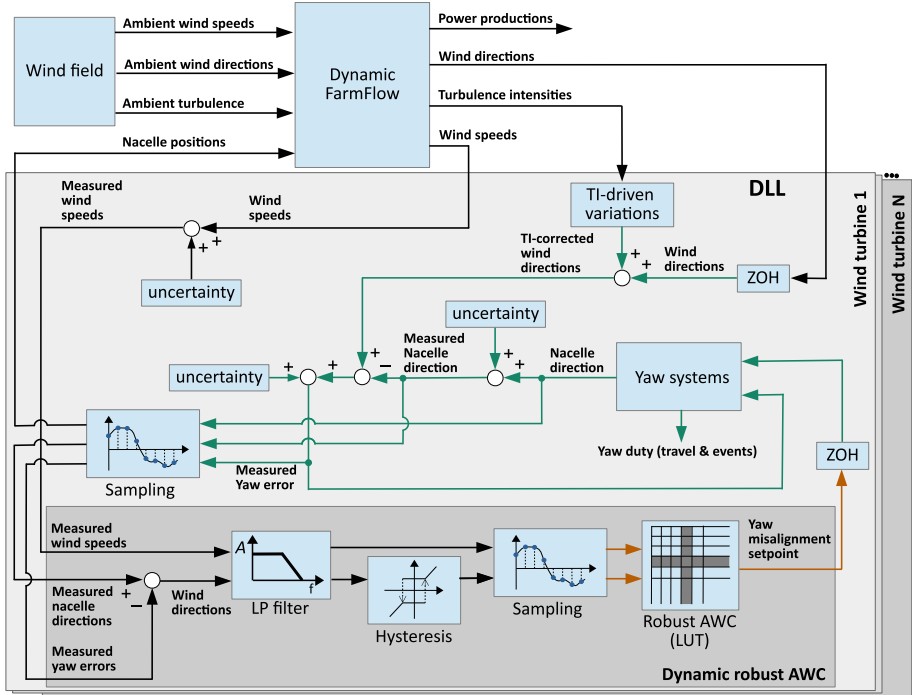

**Figure 1.** Block scheme of the simulation model

may operate at slower sampling rates (red lines). The main components are explained in more detail in the remainder of this section.

## 2.1 Wind field

A stochastic wind field generator is created that produces two-dimensional wind fields with spatiotemporal variations, enabling dynamic wind farm simulations for analysis and design of wind farm control strategies. Control-oriented wind farm simulations require much slower time scales (tens of seconds) than the aerodynamic simulations needed for evaluating wind turbine controls (which are typically in the tens to hundreds of milliseconds range). Moreover, wind farm simulations require wind fields that extent beyond the traditional duration in wind turbine simulations, usually limited to ten minutes wind fields with a spacial range of up to several hundred meters (micro scale). The size of some current wind farms extend to ten kilometres or even more (meso scale). The approach to wind field generation followed in this work is based on modelling of both micro and meso scale effects. The chosen spacial resolution of the wind fields is around two rotor diameters (2D), while the sample time is in the order of 10-30 s (roughly equal the time it takes air flow to cover a distance of 2D).

Micro scale (fast) wind variations are modelled using the Kaimal spectrum, rewritten as function of the wave number $\nu = f/U_m$, as suggested in Bossanyi (2018)

$$\frac{U_m S_{micro}(\nu)}{\sigma_m^2} = \frac{4L}{(1+6L\nu)^{\frac{5}{3}}},$$





with $f$ being the frequency, $L$ – the integral scale parameter ($L = 340.2$ m for the longitudinal wind component, and $L = 113.4$ m for the lateral wind component in accordance with IEC 61400-1-3:2005 (2005)), and $U_m$ and $\sigma_m$ being the wind velocity and its standard deviation. Writing the spectrum in this form allows the generation of long time series independent on $U_m$ or $\sigma_m$, which need not to be constant. Generated time-series are then scaled to obtain the wind realization based on the actual (slow) variation of $U_m$ and $\sigma_m$ coming from the generated meso-scale winds. This Kaimal spectrum is used for frequencies above

$10^{-3}$ Hz, i.e. time scales of 30 minutes and slower. The upper limit of the frequency range is $U_m/(2D)$, based on an assumed grid size of 2D. This implies $10^{-3}/U_m \leq \nu \leq 1/(2D)$, where finally the lower is substituted by $4.10^{-5}\ m^{-1}$ (corresponding to the maximal expected wind speed of 25 m$s^{-1}$) to remove the dependency on $U_m$ completely.

The meso-scale (slow) wind variations, in the order to 10 minutes or longer, are modelled by the auto power spectrum

$$S_{meso}(f) = a_1 f^{-5/3} + a_2 f^{-3}$$

with $a_1 = 3.10^{-4}$ and $a_2 = 3.10^{-11}$ fitted to measurement data (Larsén et al., 2013). This spectrum is representative in the frequency interval $10^{-5} \leq f \leq 10^{-3}$, i.e. approximately between 30 minutes and one day, and valid for both the longitudinal and lateral wind component.

For the complex cross power spectrum between two points in space, $r$ and $s$, the following expression is used for both the micro and the meso scale spectra (denoted below shortly as $S$) and is adopted from Sørensen et al. (2002)

$$S_{rs}(f) = \gamma(f, d_{rs}, U_0) S(f) e^{-j2\pi\ f\tau_{rs}(U_0)}, \tag{1}$$

where $d_{rs}$ is the distance between the two points, $U_0$ is the average wind velocity, $\tau_{rs}(U_0) = (\cos(\alpha_{rs}) d_{rs})/U_0$ is the time delay, i.e. the time it takes to move downstream from point $r$ to point $s$ at the average wind velocity $U_0$,

$$\gamma(f, d_{rs}, U_0) = \exp^{-c(\alpha_{rs}) d_{rs} f/U_0}$$

is the coherence function between the points $r$ and $s$, and $\alpha_{rs}$ denotes the angle between the line through the points and $r$ and

$s$ and the wind velocity vector. The advection of the airflow downstream is modelled by the exponent term in Eq. (1).

The parameter $c(\alpha_{rs})$ is the decay factor. There is no uniformity in the literature with respect to the decay factor. In this work, the decay parameter $c(\alpha_{rs}) = 5.9 - 1.8\cos(2\alpha_{rc})$, as suggested by Larsén et al. (2013), has been used.

The generation of time series from auto and cross power spectra follows the standard approach of generating frequency domain signals with amplitudes complying with the specified spectra and random phases, and applying inverse fast Fourier

transform to construct the time series (Veers, 1988).

## 2.2   Dynamic wake model

In this work a dynamic wake model is developed based on TNO's wake model FarmFlow (Bot and Kanev, 2020), a Parabolized Reynolds-averaged Navier-Stokes code with prescribed pressure gradients to calculate the flow in wind farms. Based on the rotor averaged wind speeds, the power and induced velocities are determined from measured power and thrust curves. The

pressure gradients in the near wake region are prescribed as a function of the thrust force coefficient. To this end, a database is



used containing precomputed pressure gradients obtained from a panel method with an actuator disk model in which the wake is represented by discrete constant strength vortex rings. The basic background flow is modelled by an atmospheric wind shear model based on Monin-Obukhov similarity theory. The original FarmFlow model solves stationary flow throughout the wind farm.

For the development and analysis of dynamic AWC algorithms, FarmFlow has been extended to model a quasi-dynamic flow in two-dimensional wind fields at hub height (see Sect. 2.1) with spatiotemporal variations. To this end, the wake generated by a wind turbine is propagated downstream based on the local wind direction variations on its way, thereby modelling both time delay and meandering effects. The simulation time is equal to the time step in the wind field time series. Because the traveling time of the wakes between two turbines takes much longer than the time window of a simulation period, the arriving wakes from upstream turbines need to be time synchronized with the departure of wakes in the current time window. Because the streamlines of the two-dimensional wind fields are curved and are varying in time, the trajectory of each wake needs to be corrected at the location of arrival, based on the trajectory of the undisturbed flow. After the correction, the wakes are stored in memory including the time and location of arrival. In summary, the quasi-dynamic wind farm simulation is realized as follows:

1. A simulation for the current time window of the wind field starts with the most upstream turbine and ends with the most downstream turbine, as seen from the average wind direction.

2. Before the wake simulation for a turbine starts, arriving upstream wakes at the current time instant are first determined, if any, using the wake information stored at previous time instances (Step 5).

3. From the undisturbed wind field and arriving wakes, the rotor averaged wind speed and wind direction is calculated.

4. Given the determined rotor averaged wind speed and the nacelle direction (coming from the yaw system model, see Fig. 1), the yaw misalignment angle is computed, the power and thrust values of the turbine are determined and a static wake calculation using the (stationary) wake model in FarmFlow is started for the given turbine only.

5. The wake is calculated until it hits a downstream turbine, if any, in which process the precise location of the wake is corrected for the time varying wind direction from the undisturbed wind field. The wake information is then stored in memory, including the time of arrival at the downstream turbine for use in the wake simulation of that turbine later on (performed in step 2).

6. The simulation continues with the next upstream turbine until all turbines are finished within the current time instant.

7. The input and output data for all turbines are written in an output file, which forms the interface to a dynamic link library (DLL) that implements the yaw system dynamics and the dynamic robust AWC algorithm.

8. The DLL is called, which updates the yaw position for each turbine and communicates this information back to Farm-Flow, and the next simulation step is started (step 1).

This process is repeated until the simulations for all time periods are finished.





## 2.3 Yaw model

Similarly to Bossanyi (2018), the yaw system model is simplified to a constant rate motion at $0.35°s^{-1}$ and a simple yaw controller activating the yawing motion when the LP filtered difference between the yaw misalignment and its set-point exceeds
$8°$. A second order Buttherworth LP filter with cut-off frequency of $1/60Hz$ has been used. As depicted in Fig. 1, the outputs of the yaw model are the nacelle position and the yaw duty (yaw travel and number of yaw on-off events). The inputs are the yaw misalignment set-point and the measured yaw misalignment. The later is constructed as the difference between the measured local wind direction (coming from the wake model) and the measured nacelle direction. The measured values are formed by perturbing the actual quantities with uncertainties, i.e. terms representing variability of the measurand, measurement noise and
uncertainty. These uncertain terms can be either stochastic or deterministic (e.g. measurement bias).

    As pointed out already, the yaw system model operates at faster sampling rates than the wake model. This allows to model the yaw dynamics properly to enable realistic assessment of the impact of AWC on the yaw duty. To account for the higher variability of wind direction measurements that are taken at higher sampling frequencies, additional noise is superimposed on the wind directions coming from the wake model. Besides the higher sampling frequency, there is another source of increased
noise on the wind directions, namely the turbulence intensity of the air flow impinging the turbine. Operating in the wake of other turbines, a wind turbine will measure an increased level of wind direction variations. These, together with the additional noise due to higher sampling frequency, are represented by the block named "TI-driven variations" in Fig. 1. The additional noise signal, added by this block, is generated based on the standard deviation of the wind velocity in front of the wind turbine. More specifically, denoting $\tilde{\phi}(t) = \tan \frac{v(t)}{u(t)}$ as the angle between the longitudinal ($u(t)$) and lateral ($v(t)$) components of the
wind vector, representing variations of the wind direction around its average value $\phi_0$, it can be shown using the Taylor series approximation of degree one around the point $(u = u_0, v = 0)$ that

$$\tilde{\phi}(t) \approx \frac{v(t)}{u_0}.$$

This implies that the standard deviation of the wind direction $\phi(t)$ can be expressed as $\sigma_{WD} = \sigma_v/u_0$, where $\sigma_v$ is the standard deviation of the lateral wind component. In free stream, IEC 61400-1-3:2005 (2005) recommends for $\sigma_v$ the expression $\sigma_v =$
$0.8\sigma_u$, which allows one to approximate the standard deviation of the wind direction to the turbulence intensity, namely

$$\sigma_{WD} \approx \frac{0.8\sigma_u}{u_0} = 0.8TI(u).$$

Assuming that this expression is representative for both the ambient flow as well as the wake, one can write $\sigma_{WD,amb} = 0.8TI_{amb}$ for the ambient wind direction, and $\sigma_{WD,wake} = 0.8TI_{wake}$ for that in the wake. In the simulation model, the ambient turbulence intensity, $TI_{amb}$, comes from the wind field generator, while the turbulence intensity of the wake in front of
a wind turbine, $TI_{wake}$, is calculated by the wake model FarmFlow. The purpose of the block "TI-driven variations" in Fig. 1 is to add additional noise to the higher frequency wind directions used in the yaw system model to obtain $\sigma_{WD,wake} = 0.8TI_{wake}$. This additional noise is generated using the Kaimal spectrum with standard deviation $\sigma_{WD,add}$ according to the following expression, which is a direct consequence of the derivation outlined above

$$\sigma_{WD,add} = 0.8\sqrt{\max\{0, TI_{wake}^2 - TI_{amb}^2\}}.$$





**Table 1.** Impact of wake on wind measurements and yaw motion of two commercial wind turbines

| Turbine | Inflow | wind speed | | | wind direction | | Yaw duty | |
| | | STD | mean | TI | STD | mean | travel | nr. events |
| | | $[ms^{-1}]$ | $[ms^{-1}]$ | [-] | [°] | [°] | [°] | [-] |
| T1 | free stream | 2.41 | 10.00 | 0.24 | 15.66 | 179.56 | 39.32 | 6 |
| T2 | free stream | 2.43 | 10.63 | 0.23 | 15.69 | 182.84 | 43.85 | 8 |
| T1 | free stream | 1.37 | 7.12 | 0.19 | 11.91 | 271.28 | 23.09 | 4 |
| T2 | in wake | 1.66 | 4.50 | 0.37 | 17.43 | 278.86 | 81.50 | 9 |

This results in increased variations of the measured wind direction at turbines operating in wake condition, which in turn gives rise to increased yaw motion. This fact is observed in real-life as well, as supported by the data presented in Table 1. These data are obtained using high frequency measurements of wind speed, wind direction and nacelle position on two commercial wind turbines in the 2.5MW range, located on flat terrain at a distance of around four rotor diameters. The turbines operate both in free stream for Southern winds, while the second turbine (T2) operates in the wake of the first one (T1) for Western

winds. The last row in the table shows that, in wake, T2 experiences lower wind velocity and higher turbulence intensity than the free stream turbine T1, as expected. It also shows that T2 measures increased variation in the measured wind direction and, unsurprisingly, higher yaw duty. This can be observed in the time series of LP filtered wind direction and raw nacelle position measurements for the two turbines in Fig. 2, which is for the Western wind situation. For Southern winds the responses are comparable (not shown in the Figure). These results support the concept of modelling increased wind direction variability and

yaw activity for waked turbines. Notice that the developed model cannot be easily validated by only using such SCADA data, since the reported turbulence intensities in the table are computed from measurements disturbed by the rotor. Nevertheless, the model produces comparable results in terms of yaw motion and is considered sufficient for the purpose of this work.

## 2.4 Dynamic robust AWC

The dynamic robust AWC consists of the following blocks (see Fig. 1):

– LUT containing the yaw misalignment set-points for all wind turbines in the wind farm as function of the wind speed and wind direction, obtained through stochastic program that accounts for uncertainty in a number of parameters (see Sect. 3 for more details). The optimization is performed using the conventional (stationary) FarmFlow model for a range of wind conditions to populate the LUT.

– LP filter: a second order Butterworth lowpass filter, the cutoff frequency of which will be varied to study its impact on

the power gain and yaw duty.

– Hysteresis: adding hysteresis on the (filtered) wind direction signal, centred at wind directions where the yaw misalignment set-points change sign, can reduce the yaw duty as demonstrated in earlier by Kanev (2020). The hysteresis logic





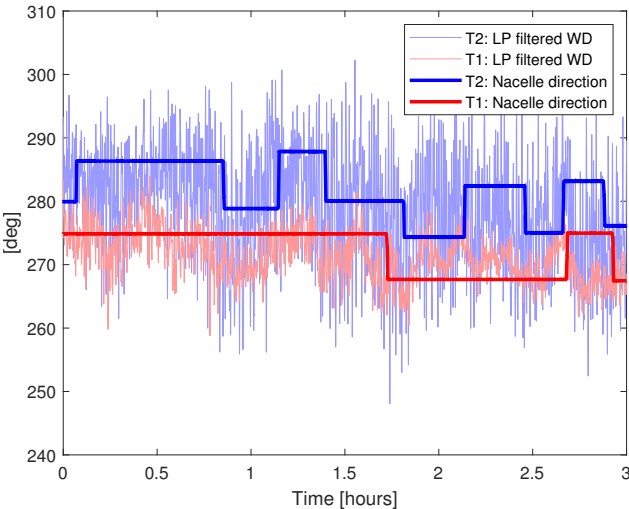

**Figure 2.** Impact of wake on wind measurements and yaw motion of two commercial wind turbines

for a given turbine is defined as

$$\mathcal{H}(b): \ \phi_{hyst}(k) = \begin{cases} \phi_{hyst}(k-1) & |\phi_{LP}(k) - \phi_c| \leq b \\ \phi_{LP}(k) & \text{otherwise} \end{cases}$$

wherein $\phi_c$ is any wind direction for which the yaw misalignment set-point for that turbine changes sign, $\phi_{LP}(k)$ is the LP filtered wind direction (input to the hysteresis), and $b$ defines the hysteresis size. Based on the findings in Kanev (2020), a wider range of hysteresis sizes are considered in the present work ($b$ up to $4°$).

  – Sampling: limits the update rate of the yaw misalignment set-points, i.e. the frequency at which they are communicated to the yaw controller implemented in the yaw system model.

The robustness of the AWC is realized through the robust design of the LUT, while the last three components of the AWC controller in the list above represent the dynamic adaptation algorithm. The robust design through stochastic programming will be discussed in Sect. 3. The quantities to which robustness is to be achieved are selected through uncertainty quantification in Sect. 3.2. The selection of dynamic adaptation algorithm parameters (LP filter cut-off frequency, hysteresis size and sample time) to optimize the balance between power gain and yaw duty is topic of Sect. 4.

**3   Robust AWC optimization**

The conventional approach to (nominal) AWC design involves the synthesis of LUT containing yaw misalignment set-points for the wind turbines in the farm, optimized for a range of input conditions. These include at least the wind direction (Kanev et al., 2018), but sometimes also the wind speed or other atmospheric conditions such as the turbulence intensity (Bossanyi, 2018;





**Table 2.** Uncertain parameters considered in the uncertainty quantification framework, their assumed ranges and stochastic modelling

| Parameter | Uncertainty type | Range | PDF |
|---|---|---|---|
| Wind direction | Variability and measurement uncertainty | $[-10, 10]^\circ$ | Normal $(\mu = \phi_{LUT}, \sigma = 4.25^\circ)$ |
| Wind speed | Variability | $[4, 12]\ ms^{-1}$ | Weibull $(k = 2.24, A = 9.3)$ |
| Turbulence intensity | Variability | $[4, 20]\ \%$ | Weibull $(k = 3, A = 0.073)$ |
| Yaw error | Variability and measurement uncertainty | $[-10, 10]^\circ$ | Laplace $(\mu = 0, \nu = 5^\circ)$ |
| Wind shear exponent | Variability and model uncertainty | $[0.04, 0.20]$ | Normal $(\mu = 0.12, \sigma = 0.05)$ |
| Air density | Variability | $[1.18, 1.38]\ kgm^{-3}$ | Normal $(\sigma = 0.015)$ |
| Yawed power loss exp. | Model uncertainty | $[1.3, 2.3]$ | biased inverse Gaussian $(\mu = 0.52, \lambda = 8)$ |
| Thrust curve variation | Model uncertainty | $[-10, 10]\ \%$ | Normal $(\mu = C_{t,nom}, \sigma = 10/3\ \%)$ |
| Power curve variation | Model uncertainty | $[-5, 5]\ \%$ | Normal $(\mu = C_{p,nom}, \sigma = 5/3\ \%)$ |

Doekemeijer et al., 2021). In this work, the yaw set-points will depend only on the wind direction, while the influence of other
parameters on their optimal choice will be considered through stochastic uncertainties in a robust design setting. Nevertheless,
some dependency on the wind speeds is inevitable as AWC should not be active above the rated power production of the farm.
In other words, above a certain wind velocity ($12\ ms^{-1}$ used here) the yaw set-points should be phased out. Same holds for
too low wind velocities (here $4\ ms^{-1}$), where large misalignment may interfere with the start-up process.

This section considers the synthesis of robust AWC in which a number of parameters is treated as uncertain. These consist
of quantities related to model parameter uncertainty, measurement uncertainty and input variability, all treated in a unified
stochastic framework using PDFs (Sect. 3.1). The impact of these uncertainties on the optimal yaw set-points is quantified
through forward uncertainty propagation (Sect. 3.2). To this end, the PDFs are sampled and the underlying simulations are
performed using the conventional (static) FarmFlow wake model. The robust AWC design is performed by solving a stochastic
programming problem through discretization of the probability distributions to arrive at a finite number of scenarios (Sect. 3.3).

## 3.1 Uncertainty modelling

The uncertainties considered in this work are listed in Table 2, together with their type, assumed range and the PDF used
for their modelling. For instance, the wind direction is required in the AWC algorithm as input, and since it is derived from
the measured nacelle direction and yaw error, it will be subjected to measurement uncertainty. Next to that, the actual wind
direction will vary with respect to the one entering the LUT, denoted in the table as $\phi_{LUT}$, due to the applied signal processing
in the AWC algorithm (see Fig. 1). The table indicates that the measurement uncertainty and variability are modelled together
with a normal distribution with PDF

$$p_{\text{normal}}(x, \mu, \sigma) = \frac{1}{\sigma\sqrt{2\pi}} e^{-\frac{(x-\mu)^2}{2\sigma^2}}$$

with standard deviation $\sigma = 4.25$ and mean value equal to the wind direction entering the LUT, $\mu = \phi_{LUT}$. Notice that the
value for the standard deviation is difficult to compute accurately because it needs to reflect the difference between the wind





direction variations responsible for the wake meandering, say $\phi_{meand}$, and $\phi_{LUT}$, and the former cannot be exactly specified. The value of 4.25, used here, is determined by using the high frequency measurement data, discussed in Sect. 2.3. To this end, 5s LP filtering is applied to the high-frequency wind direction measurements to represent $\phi_{meand}$, and 20s LP filtering to model $\phi_{LUT}$, and a normal distribution has been fitted to the difference between the two signals, wherein the standard deviation has slightly been increased to account for the impact of measurement uncertainty on the lower frequency signal $\phi_{LUT}$. Notice that the so determined value does not differ much from that used in earlier research (Simley et al., 2020; Rott et al., 2018; Quick et al., 2020).

The wind speed, as already mentioned, will not be used as input to the LUT (other than for defining the wind speed region in which intentional yaw misalignment will be applied). Therefore, only variability of the wind speed signal is considered, hence neglecting the effect of measurement uncertainty which is much smaller in our setting. For its variability, a standard Weibull distribution of the form

$$p_{\text{Weibull}}(x, k, A) = \frac{k}{A}\left(\frac{x}{A}\right)^{k-1} e^{-(x/A)^k}$$

is used, and the values for its parameters $k$ and $A$, reported in Table 2, representative for a commercial offshore wind farm in the North sea.

To determine a suitable PDF for the turbulence intensity, twenty months of 10 min statistical data has been used from a met mast located offshore in free stream. From these data, a histogram has been constructed for the turbulence intensity for wind velocities in the range of 4-12 $ms^{-1}$ since, as mentioned already, in this study AWC is not active outside this interval. The histogram is depicted in Fig. 3, and shows similarity with the Weibull distribution. A fit delivered parameters $k = 3$, and $A = 0.073$, for which the fitted PDF looks quite reasonable, even though the probability of values above 0.15 is underestimated.

The PDF parametrizations for yaw error and wind shear exponent have been adopted from Quick et al. (2020). The Laplace distribution, assumed for the yaw error uncertainty is defined as

$$p_{\text{Laplace}}(x, \mu, \nu) = \frac{1}{2\nu} e^{-|x-\mu|/\nu}.$$

The uncertainty on the air density has been based on the results in Ulazia et al. (2019), where a study is carried out on the variability of the air density offshore. The statistical indicators, reported there, are a first quartile value of 1.21 and third quartile value of 1.25. The resulting interquartile range of $IQR = 0.02$ is then used here to determine the standard deviation for which the normal distribution exhibits the same $IQR$, i.e. $\sigma = IQR/1.35 \approx 0.015$.

The parameter denoted as "yawed power loss exp." in Table 2 refers to the exponent $a$ in the power reduction factor $\cos(\beta)^a$ by which the power production of a non-yawed turbine is scaled to model the production of a turbine operating at yaw misalignment angle of $\beta$. There is a large variety of values for $a$ in the literature, ranging between 1.3 and 2.3. In Fleming et al. (2017) a value ot 1.44 has been fitted to field measurements. Similar value has been derived for another commercial wind turbine, used in the study Doekemeijer et al. (2021). For this reason, a skewed PDF is selected here, defined as

$$p_{\text{biG}}(x, \mu, \lambda) = \sqrt{\frac{\lambda}{2\pi(x-1)^3}} e^{-\lambda(x-1-\mu)^2/(2\mu^2(x-1))}$$



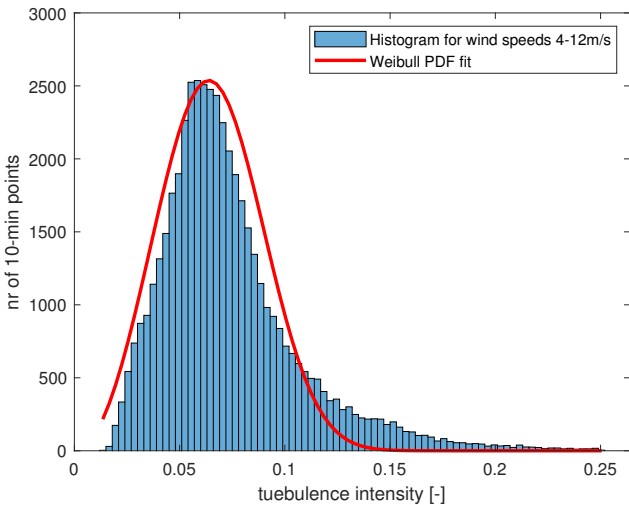

**Figure 3.** Histogram of turbulence intensity computed using 20 months of wind measurements on an offshore met mast, and a Weibull PDF fit. Only data corresponding to wind speeds in the interval [4,12] $ms^{-1}$ have been used in constructing the histogram

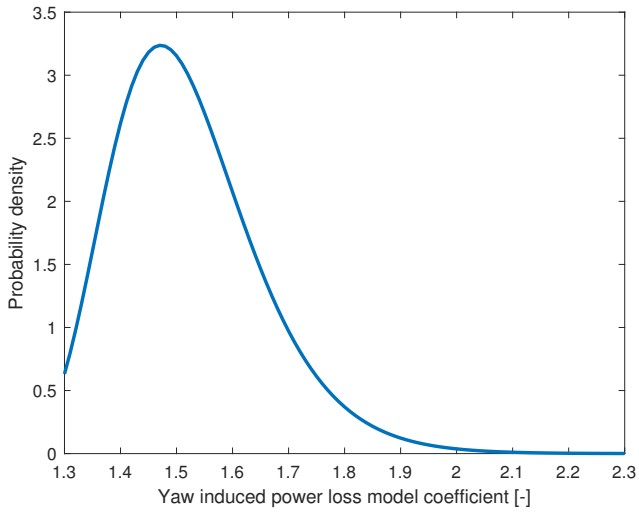

**Figure 4.** PDF used for representing model uncertainty in the yaw-induced power loss coefficient

with highest probabilities clustered at the lower side of the range, see Fig. 4. The selected PDF is such that for $p_{\mathbf{biG}}(x+1, \mu, \lambda)$ it becomes equivalent to the inverse Gaussian distribution, and is therefore referred to as "biased inverse Gaussian" in Table 2.

The mode of the PDF, at 1.47, aligns quite well with mentioned field results.

Finally, the uncertainties on the thrust and power curves are modelled relative changes (percentages) of their nominal curves, i.e. $\pm 5\%$ uncertainty range on the power curve, and $\pm 10\%$ on the thrust curve. The modification of the power curve is realized



through scaling of the wind speed to ensure that rated power remains unchanged. Normal distributions are then assumed, centred at nominal values, and with standard deviations such that $\pm 3\sigma$ coincide the selected uncertainty ranges.

Now that the possible sources of uncertainty are described and modelled, the next section continues with analysing the impact of these on the performance of the AWC algorithm.

## 3.2 Uncertainty quantification

The uncertainty quantification analysis has been carried out with the purpose to identify the uncertainties with the most significant impact on the performance of the AWC algorithm, measured in terms of optimal yaw misalignment set-points and power

gain. The most dominant uncertainties are then to be considered in the robust AWC design framework, discussed in the next section. The reason to look for reduction in the number of uncertainties in the robust optimization is to lower the computational complexity to a manageable level.

The analysis in this section is performed as follows. First, a number of equally distant samples is selected for each uncertain quantity within its assumed range (see Table 2). Next, for each uncertainty sample, the yaw misalignment set-points were

optimized for a row of five 3 MW wind turbines with 90 m rotor diameter for the following three cases

**Case 1** inter-turbine spacing of five rotor diameters (5D) and wind direction aligned with the row

**Case 2** inter-turbine spacing of 7D and wind direction aligned with the row

**Case 3** inter-turbine spacing of 7D and wind direction an angle of $4°$ with respect to the row

The yaw misalignment angles are optimized for one uncertainty parameter at a time, keeping all other parameters at their

nominal values. More precisely, let $p_i$ be a given parameter from the list in Table 2, $\mathcal{U}_i$ be the corresponding uncertainty range, and $\mathcal{D}_i(p_i)$ – its PDF. For a given sample $p_i^{(r)} \in \mathcal{U}_i$ of the parameter $p_i$, and keeping the remaining parameters at their nominal values (i.e. $p_j = p_j^{(nom)} \in \mathcal{U}_j$, $j \neq i$), conventional (non-robust) AWC design solves the optimization problem of finding the vector of best yaw misalignment set-points $\gamma = [\gamma_1, \ldots, \gamma_N]$, $N$ being the number of turbines, with respect to the total power production of the wind farm, i.e.

$$\gamma_{det}(p_i^{(r)}) = \arg\min_{\gamma} \sum_{t=1}^{N} P_t\left(\gamma | p_i = p_i^{(r)}, p_j = p_j^{(nom)}, j \neq i\right).$$

Notice that each individual turbine's power production, $P_t$, may depend on the yaw misalignments of other turbines through the wake effects. The optimal power gain for sample $p_i^{(r)}$ is then defined as

$$\delta P_{opt}(p_i^{(r)}) = \left(\sum_{t=1}^{N} P_t\left(\gamma_{det}(p_r) | p_i = p_i^{(r)}, p_j = p_j^{(nom)}, j \neq i\right)\right) \bigg/ \left(\sum_{t=1}^{N} P_t\left(\gamma = 0 | p_i = p_i^{(r)}, p_j = p_j^{(nom)}, j \neq i\right)\right).$$

Table 3 gives for each parameter $p_i$ its uncertainty set $\mathcal{U}_i$, the selected step size in the uncertainty sampling (resulting in

samples $p_i^{(r)}$, $r = 1, 2, \ldots$ for which the yaw misalignments are optimized), as well as the nominal values of the parameters ($p_j^{(nom)}$). The resulting optimal yaw misalignment angles $\gamma_{det}(p_i^{(r)})$ are statistically summarized with the box plots in Fig. 5.





**Table 3.** Sampling of the uncertainties for the purpose of uncertainty quantification

| Parameter | Range | Step size | Nominal value |
|---|---|---|---|
| Wind direction | $[-10, 10]^\circ$ | $1^\circ$ | $0^\circ$(Cases 1,2), $4^\circ$(Case3) |
| Wind speed | $[4, 12]\ ms^{-1}$ | $1\ ms^{-1}$ | $8\ ms^{-1}$ |
| Turbulence intensity | $[4, 20]\ \%$ | $1\ \%$ | $7\ \%$ |
| Yaw error | $[-10, 10]^\circ$ | $1^\circ$ | $0^\circ$ |
| Wind shear exponent | $[0.04, 0.20]$ | $0.02$ | $0.09$ |
| Air density | $[1.18, 1.38]\ kgm^{-3}$ | $0.02\ kgm^{-3}$ | $1.225\ kgm^{-3}$ |
| Yawed power loss exp. | $[1.3, 2.3]$ | $0.1$ | $1.43$ |
| Thrust curve variation | $[-10, 10]\ \%$ | $2\ \%$ | $0$ |
| Power curve variation | $[-5, 5]\ \%$ | $1\ \%$ | $0$ |

In the figure, three box plots per parameter are depicted, corresponding to the considered three cases. Each box plot gives the minimum and the maximum value (upper and lower line segment), the first and third quartile (lower and upper sides of the boxes), and the median (line segment inside the box).

The following observations can be made from the figure

– Variations in the power coefficient and the air density have no impact on the optimal yaw misalignment set-points, which is expected as these parameters have little to no influence on the wake deficits behind the turbines

– Wind direction variability has by far the largest impact on the optimal yaw misalignment set-points which, of course, is due to their very pronounced influence on the wake locations with respect to downstream turbines. Clearly, this parameter
is the most important one to consider in a robust AWC setting.

– Other quantities, variations of which lead to significant changes in the optimal yaw set-points, are the wind speed, yaw error, turbulence intensity, and wind shear. Notice that for some turbulence intensity and wind speed cases the minimum yaw misalignment values are equal to zero, which occur for uncertainty samples around the edges of their ranges of variation. For measurable quantities (such as the wind speed), such values can better be excluded from the
robust optimization when possible and be used instead to deactivate AWC.

– Variations in the thrust curve and the yaw-induced power loss exponent have generally limited impact on the optimal yaw set-points, which suggests that they could be left out from the robust optimization.

Next, the sensitivity of the power gain by AWC to variations in the uncertain parameters is considered. The reason for doing that is that it might happen that the variations in a given parameter lead to large changes in the corresponding optimal yaw set-
points, but limited variation in the power gains, in which case inclusion of the parameter in question into the robust AWC design will also be unnecessary. Therefore, simulations have performed for the selected uncertainty samples, as described above, $p_i^{(r)}$,



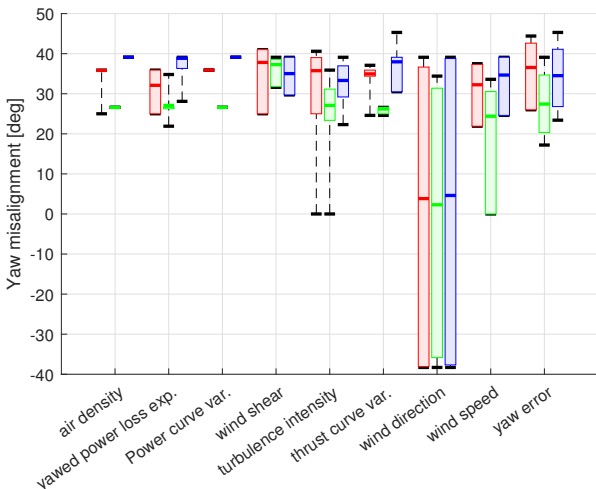

**Figure 5.** Box plots for the optimal yaw misalignment angles of the most upstream turbine, evaluated for the considered parameters and setup cases

$r = 1, 2, \ldots$, but with the turbines' yaw misalignment set-points fixed over the whole uncertainty interval for a given parameter. The power gain sensitivity analysis is carried out as follows. For each parameter, its most probable sample (modal value), $p_i^{(mod)}$, is selected and the yaw misalignment angles are optimized for that value, while keeping all other parameters at their

nominal value ($p_j = p_j^{(nom)}$, $j \neq i$), i.e.

$$\gamma_{fix}(p_i^{(mod)}) = \arg\min_{\gamma} \sum_{t=1}^{N} P_t \left( \gamma | p_i = p_i^{(mod)}, p_j = p_j^{(nom)}, j \neq i \right).$$

For each parameter sample ($p_i^r$, $r = 1, 2, \ldots$), the power ratio is then computed as

$$\delta P_{fix}(p_i^{(r)}) = \left( \sum_{t=1}^{N} P_t \left( \gamma_{fix}(p_i^{(mod)}) | p_i = p_i^{(r)}, p_j = p_j^{(nom)}, j \neq i \right) \right) \Bigg/ \left( \sum_{t=1}^{N} P_t \left( \gamma = 0 | p_i = p_i^{(r)}, p_j = p_j^{(nom)}, j \neq i \right) \right).$$

By optimizing for one single uncertainty value ($p_i^{(mod)}$), evaluating the power gain with the resulting (fixed) yaw set-points

($\gamma_{fix}(p_i^{(mod)})$) for the whole uncertainty range ($\delta P_{fix}(p_i^{(r)})$), and comparing these to the optimized power gains for each single uncertainty sample separately ($\delta P_{opt}(p_i^{(r)})$), one gets insight into the maximum amount of power gain improvement achievable by means of robust AWC. If there is little to no difference between $\delta P_{fix}$ and $\delta P_{opt}$ for a given parameter, then it can be left out from the robust design even if it has significant impact on the optimal yaw set-points. The difference ($\delta P_{opt} - \delta P_{fix}$) is depicted statistically with the box plots in Fig. 6. A value of 0.05 in the figure, indicates that for some uncertainty realization

$p_i^{(r)}$ the optimal power gain $\delta P_{opt}$ is 5% higher the power ratio $\delta P_{fix}$ for fixed yaw set-points. Therefore, values close to zero in the figure indicate that the uncertainty on the corresponding parameter is not important to include in the robust AWC design because that will not lead to a significant improvement in the power gain compared to the case when the AWC design



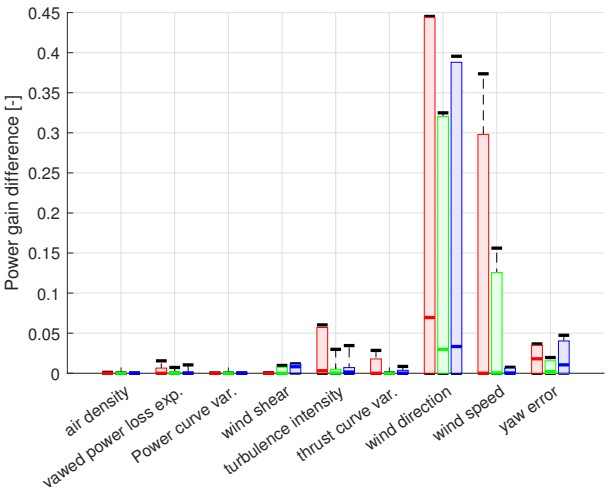

**Figure 6.** Box plots for the difference between the power gains obtained for the considered uncertainty samples with fixed (optimized for the model value of the uncertain parameter) and varying (optimized for each uncertainty sample individually) yaw misalignment set-points

is performed with the parameter kept at its modal value. This is the case for the first four parameters depicted in Fig. 6 (air density, yaw-induced power loss exponent, power curve, and wind shear), and to a lesser extend for the sixth one (thrust curve).

It is interesting to observe that wind shear can now be excluded from the robust optimization, while it passed the first relevance test based on the optimal yaw misalignment set-points.

In summary, the conclusion is that the uncertainty on the following parameters is to be considered in the robust AWC design framework: wind direction, wind speed, yaw error and turbulence intensity.

### 3.3 Robust design and stationary analysis

As discussed in the previous section, the robust AWC optimization will account for uncertainties in the wind direction, wind speed, yaw error and turbulence intensity, with the uncertainty modelled stochastically by means of PDFs (see Sect. 3.1). Then, for a given stationary wind direction $\phi_{LUT}$, the robust AWC design problem, will ideally require the solution to the following stochastic programming problem

$$\gamma_{rob}(\phi_{LUT}) = \arg\min_{\gamma} \int_{p \in \mathcal{U}} \sum_{t=1}^{N} P_t\left(\gamma|p, \phi_{LUT}\right) \mathcal{D}(p) dp, \tag{2}$$

wherein the four element vector $p$ represents the considered uncertain parameters, the set $\mathcal{U}$ defines their range of variation (in accordance with Table 2), and $\mathcal{D}$ – their joint PDF which, due to assumed dependency between the parameters, equals here the product of the four individual PDFs. This optimization problem needs to be solved separately for all relevant wind directions to populate the whole LUT with the optimized yaw set-points, which represents the robust AWC.





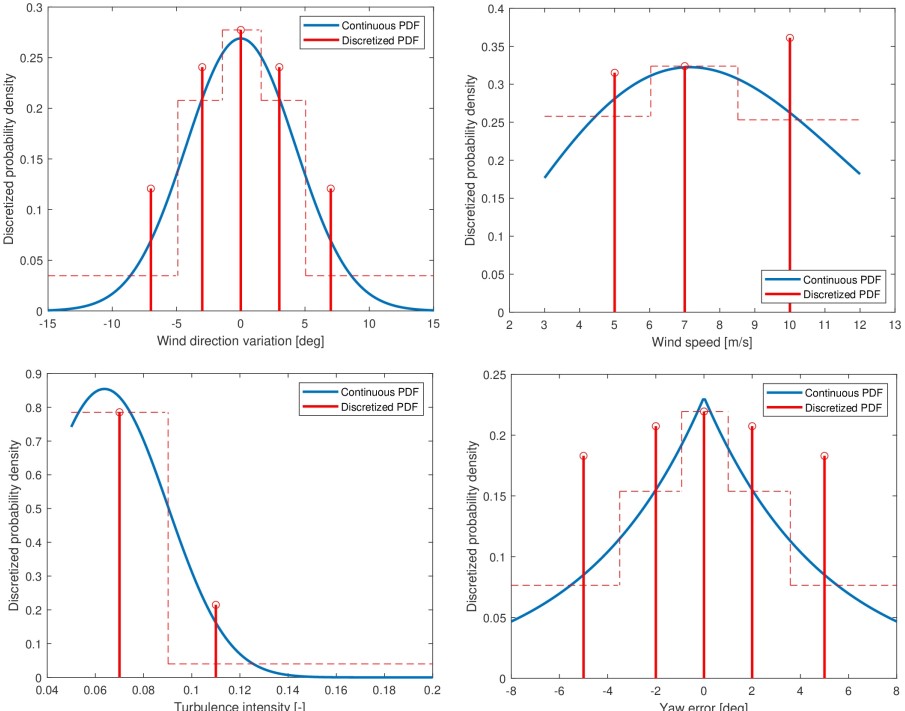

**Figure 7.** PDFs for the uncertain parameters considered for robust optimization

To solve this problem numerically, the continuous PDFs are discretized. This is done by selecting a number of discrete
samples for each uncertain parameter and calculating the probability for a given sample through integration of the continuous
PDF over the interval that corresponds to this sample (see Fig. 7). In this discretization process, an attempt has been made to
limit the total number of samples as much as possible while still trying to reasonably approximate the PDFs. Since uncertainty
on the wind direction was shown to have the largest impact on AWC, it was discretized using more points (five) than for the
wind speed and turbulence intensity PDFs (for which, respectively, three and two points have been used). Due to the symmetry
in the PDF for the yaw error, the number of points used there is also five, giving a total number of $150 \ (= 5 \times 5 \times 3 \times 2)$
joint uncertainty samples. Due to the computational complexity of the FarmFlow wake model, with the simulations run on
100 cores in this study, it was beneficial to reduce the number of cases even further. This was done by removing all parameter
combinations for which the joint cumulative probability distribution function is lower than 0.05, resulting in a final number of
121 samples. Denoting $\mathcal{U}_d$ as the set containing these 121 uncertainty samples for the vector parameter $p$, and $\mathcal{D}_d(p)$ as the
discretized joint PDF, the initial robust AWC optimization problem in Eq. (2) is approximated as follows

$$\gamma_{rob}(\phi_{LUT}) = \arg\min_{\gamma} \sum_{p \in \mathcal{U}_d} \sum_{t=1}^{N} P_t(\gamma | p, \phi_{LUT}) \mathcal{D}_d(p), \qquad (3)$$

which constitutes the optimization problem considered in this work for synthesizing robust AWC. For solving the problem
numerically, a modified pattern search optimization algorithm is used in which only decent directions are evaluated in order to



save computational effort. For the same purpose, the number of optimization variables is limited to the yaw set-points of the

two most upstream turbines in each downstream oriented row of turbines. The yaw set-points for the remaining turbines in the

row are linearly decreased between the second turbine and the last one, which has zero yaw misalignment set-point.

     To exemplify the robust AWC design, consider the single-row layout consisting of five 3 MW wind turbines, with 90 m rotor

diameter, separated at a distance of 7D (equivalent to the setup in Cases 2 and 3 in Sect. 3.2). The robust optimization problem

in Eq. (3) is solved for wind directions ranging from 248° to 292° at a step of 1°, an interval centred around the row orientation

of 270°. In addition to that, a nominal AWC optimization is performed for the nominal values of the uncertainties $p^{(nom)} \in \mathcal{U}$,

$$
\gamma_{nom}(\phi_{LUT}) = \arg\min_{\gamma} \sum_{t=1}^{N} P_t\left(\gamma | p^{(nom)}, \phi_{LUT}\right), \tag{4}
$$

while evaluating the robust power gain, $\delta P(\gamma_{nom}, \phi_{LUT})$, including the uncertainties. The robust power gain for given yaw

misalignment set-points $\gamma$ and wind direction $\phi_{LUT}$ is computed using the discretized joint PDF

$$
\delta P_{rob}(\gamma, \phi_{LUT}) = \left( \sum_{p \in \mathcal{U}_d} \sum_{t=1}^{N} P_t\left(\gamma | p, \phi_{LUT}\right) \mathcal{D}_d(p) \right) \bigg/ \left( \sum_{p \in \mathcal{U}_d} \sum_{t=1}^{N} P_t\left(0 | p, \phi_{LUT}\right) \mathcal{D}_d(p) \right).
$$

The results are summarized in Fig. 8 and 9, depicting the first turbine's optimized yaw misalignment set-point and power gain

for the complete turbine array, respectively, for the considered range of wind directions. The results show that the robust yaw

misalignments exhibit lower maximum values, extend over a larger interval of wind directions and are smoother. This is ex-

pected to have a positive effect on the yaw duty, which will be evaluated with dynamic simulations in Sect. 4. In terms of robust

power gain under robust AWC, $\delta P_{rob}(\gamma_{rob}, \phi_{LUT})$ (blue curve in Fig. 9), as compared to a nominal AWC, $\delta P_{rob}(\gamma_{nom}, \phi_{LUT})$

(black curve in 9), however, the improvement by robust AWC is quite limited. The red curve in Fig. 9 gives the nominal power

gain (excluding uncertainties) under nominal AWC.

## 4 Dynamic adaptation algorithm optimization

Now that the robust AWC optimization has been discussed in the previous section, the focus here is on the optimal selection

of the parameters of the dynamics adaptation algorithm, described in Sect. 2.4. The single-row layout considered in Sect. 3.2

and 3.3 is considered sufficient for this purpose, while a more realistic assessment will be performed using a full-scale wind

farm model in Sect. 5. For the simulations here, wind field time series have been generated using the approach in Sect. 2.1. The

wind field has a duration of 6 h and sample time of 10 s, and average turbulence intensity of 7%. The simulations have been

carried out with different combinations of dynamic adaptation parameters (see Sect. 2.4) from the sets:

- LP filter time constant: 20, 30, 45, 60, 120, 300, 600 s. Notice that, since the sample time is 10 s, time constant of 20 s

460       implies no filtering at all as the filter cut-off frequency then coincides with the Nyquist frequency.

- hysteresis size: 0,1,2,3,4°

- sample time: 10, 20, 30, 40, 50, 60, 120, 300, 600 s

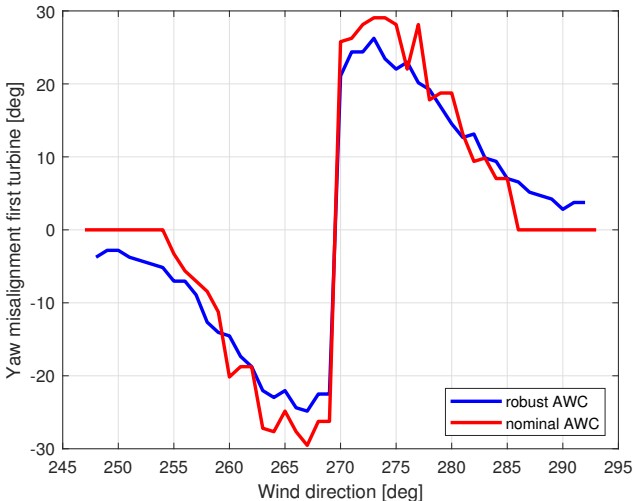

**Figure 8.** Yaw misalignment angles for the first turbine in the row

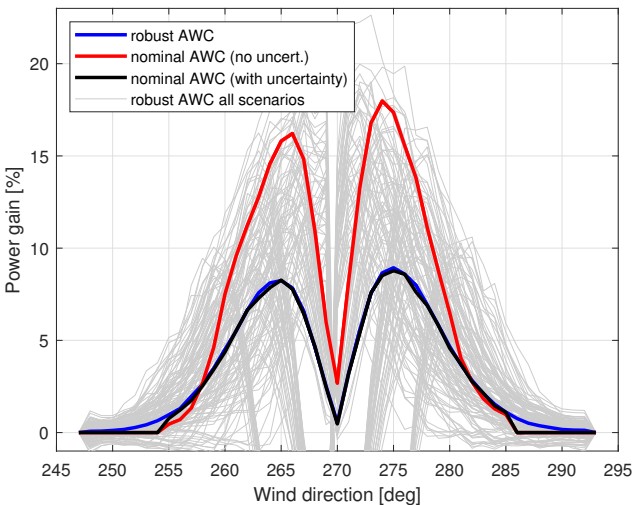

**Figure 9.** Power gain by nominal and robust AWC for a row of five turbines

Each combination of parameters is simulated twice, namely in with LUTs containing yaw misalignment set-points from nominal and robust AWC optimizations, as discussed in Sect. 3.3. Adding a reference simulation scenario without AWC results in

a total of 631 simulations ($= 7 \times 5 \times 9 \times 2 + 1$). In the sequel, the notation $DynPars(a, b, c)$ will be used to indicate a given combination of these parameters, with $a$ being the LP filter cut-off frequency, $b$ – the size parameter for the hysteresis, and $c$ – the AWC output sample time.

To exemplify the yaw motion for a few dynamic adaptation parameter scenarios, Fig. 10 is provided. Both in the left and right-hand plots, the light grey lines represent the wind direction variations at the first wind turbine in the row, while the solid





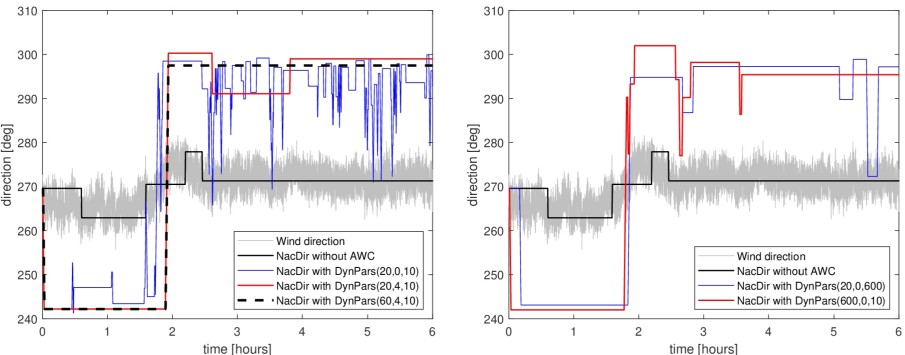

**Figure 10.** Yaw motion with AWC under different dynamic adaptation parameters

black lines give the nacelle direction (abbreviated as NacDir in the plots) in the situation with AWC inactive. The remaining
lines depict the nacelle direction with dynamic robust AWC for selected dynamic adaptation parameters. More specifically,
on the left-hand side plot, the blue line represents the yaw position with instant adaptation of the yaw set-points, and the
dashed black line – with the optimized adaptation parameters ($DynPars(60, 4, 10)$), the choice of which is explained below.
The red line in the plot on the left corresponds to the situation with maximum hysteresis size ($DynPars(20, 4, 10)$). In the

right hand side figure, the blue and red plots give the results with maximum AWC sampling time ($DynPars(20, 0, 600)$) and
maximum LP filter time constant ($DynPars(600, 0, 10)$). It can be concluded from the plots that the hysteresis size has the
most pronounced impact on the yaw duty (red and dashed black lines to the left). Long AWC sampling time (blue line in the
right plot) does reduce the yaw excursions, but also results in the yaw misalignment angles being held constant at possibly
suboptimal values for long periods of time, which may detrimental for the power gain. Long LP filter time constant appears to

result in some seemingly unnecessary yaw excursions during changes in the set-points.

The following key performance indicators (KIPs) have been evaluated for each simulation:

– energy gain: the relative increase of the wind farm energy production achieved by AWC

– average yaw travel: the amount of angular displacement travelled by all wind turbines' nacelles on the average

– worst-case yaw travel: the highest amount of angular displacement travelled by any nacelle

– average number of yaw events: the amount of start/stop yaw actions performed by all nacelles on the average

– worst-case number of yaw events: the highest amount of start/stop yaw actions performed by any nacelle

– power gain per unit yaw travel increase: relative wind farm power production gain by AWC divided by the relative
increase of average yaw travel due to AWC

– power gain per unit yaw events increase: relative wind farm power production gain by AWC divided by the relative

increase of average yaw number of yaw events due to AWC





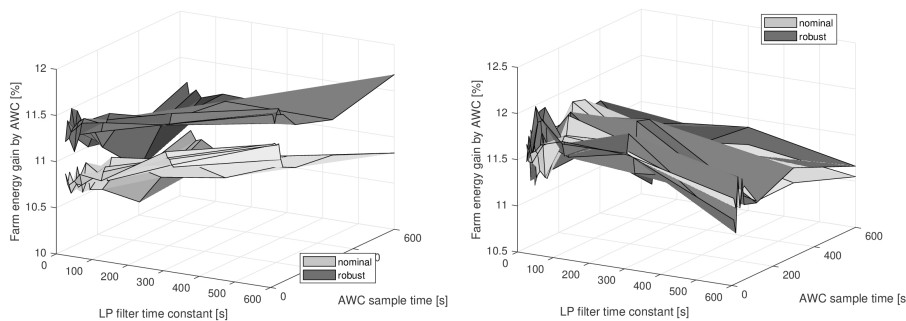

**Figure 11.** Energy gain by dynamic robust AWC and dynamic nominal AWC without hysteresis (left) and with 4°hysteresis (right)

The KPIs are depicted in the surface plots in Fig. 11 (energy gain) and Fig. 12 (remaining KPIs).

Figure 11 provides comparison between the energy gains achieved by robust and nominal AWC for different adaptation parameters. The left-hand side plot corresponds to the situation with no hysteresis ($DynPars(a, 0, c)$), while the right-hand side one is for the maximal considered hysteresis size ($DynPars(a, 4, c)$). It can be observed that, without hysteresis (left plot), the energy gain by robust AWC is higher than that with nominal AWC, and the improvement remains consistent over the whole range of adaptation parameters considered. Interestingly, the energy gain slightly increases for higher LP filter time constants, which is probably due to the stochasticity in the simulated wind field: too fast an adaptation results in the yaw misalignments trying to follow local variations of the wind direction, giving suboptimal AWC performance in terms of energy gain. This effect is strongly reduced when the largest hysteresis is in place, as seen in the right-hand side plot in Fig. 11. The local wind direction variations falling within the ±4° hysteresis zone will not appear in the yaw set-points any more, so any further LP filtering does not improve the energy gain. In fact, in that case a slight decrease in the energy gain is observed with increasing the LP filter time constant, which is attributed to the increased delay and decreased alertness of the yaw set-points to global wind direction changes.

Another interesting observation from Fig. 11 is that the case with large hysteresis gives rise to higher overall energy gain, while at the same time difference between nominal and robust AWC gets smaller.

Next, the KPIs related to yaw duty are discussed using the plots in Fig. 12, where they are expressed relatively with respect to the reference case without AWC. With respect to the four considered measures for yaw duty (yaw travel and number of start/stop events, each expressed either as an average or as a worst-case over the different turbines), the general conclusion can be drawn that increasing the hysteresis size significantly reduces the yaw duty, where huge reductions are observed for the cases with lower LP filter time constants and AWC sample times (i.e. for faster adaptation strategies). With the largest hysteresis sizes considered, the yaw duty is lowest and practically independent on the remaining adaptation parameters. This is a welcome result since, as explained above, these hysteresis sizes also turned out to improve the energy gain. Because of that, the power gain per unit yaw travel (or yaw event), depicted in the two plots on the bottom of Fig. 12, is highest for the largest considered hysteresis sizes.





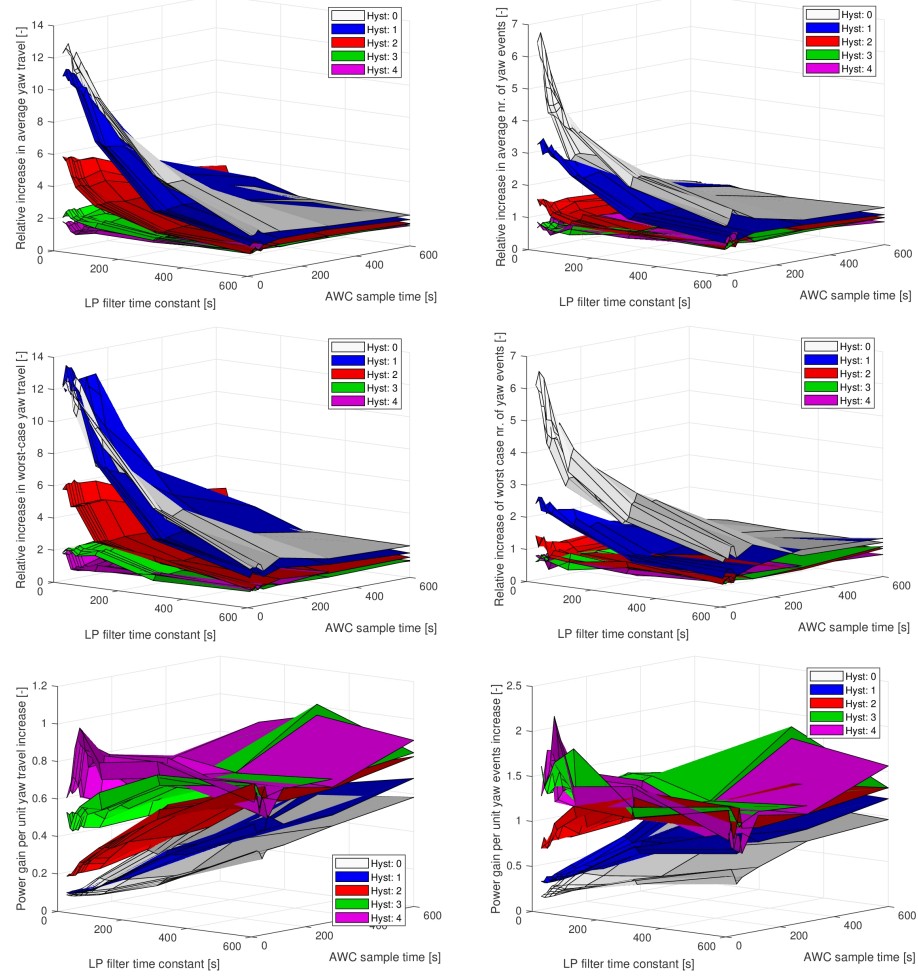

**Figure 12.** KPIs for the simulations with dynamic robust AWC: average and worst-case yaw travel, average and worst-case number of yaw start/stop events, and power gain per unity yaw travel and per unity yaw start/stop events

In summary, it can be concluded that the following adaptation parameter choices deliver best results in terms of balancing the power gain and the yaw duty, which is probably best captured by the KPIs power gain per unit yaw travel increase and power gain per unit yaw events increase:

- hysteresis size: 4 °. This popped up as the most effective parameter to reduce yaw duty and increase power gain. With the highest considered hysteresis in place, the remaining two parameters have only limited impact on the AWC performance.

- LP filter time constant: 60 s.

- sample time: 10 s.



With these adaptation parameters, the following results are achieved by the optimized dynamic robust AWC algorithm in terms of energy gain and yaw duty for the considered case study, all expressed relatively with respect to reference case (AWC-free):

– energy gain: relative increase of 12% in the wind farm energy production

   – average yaw travel: relative increase of just 1.13 (i.e. 13% increase), which is seems quite acceptable given the fact the AWC requires the nacelle to travel substantially between positive and negative offsets as the wind direction changes.

   – worst-case yaw travel: relative change of 0.95 (i.e. 5% reduction) in worst-case yaw travel. Since in the reference case it is the last turbine in the row that gets worst-case yaw travel in the simulation, this result shows that it remains higher

than the yaw travel of the first four turbines even under dynamic robust AWC.

   – average number of yaw events: relative change of 0.51 (i.e. 49% reduction) in average number of start/stop yaw actions. Having significantly less start/stop events in the reference case than with AWC might first seem couter-intuitive, but does happen. The reason for this is the negative slope in the yaw misalignment set-points to the right of their maximum value (and to the left of their minimum value), see Fig. 8, which has a damping effect on the yaw motion. It can be observed,

for instance, in the left-hand side plot Fig. 10 for the first turbine in the row (compare the solid black line with the dashed black line). For the remaining turbines (not plotted here) this effect is even more pronounced as they are yawed more often in the reference case.

   – worst-case number of yaw events: relative change of 0.67 (i.e. 33% reduction) in worst-case number of yaw events.

Altogether, it can be concluded that the results are rather positive with dynamic robust AWC achieving high energy gain and
overall reduction in the number of yaw events, at quite limited negative impact on the average yaw travel.

## 5  Case study

In this section, the dynamic robust AWC approach is evaluated on a model of an existing full-scale offshore wind farm, namely the Offshore Windpark Egmond aan Zee (OWEZ). The layout of the wind farm OWEZ is given in Fig. 13 with the rotor diameter ($D = 90m$) as unit, and where the rotor diameters (the line segments) have been drawn 50% larger for more clarity.
The wind farm is located at a distance of around 14 km off the Dutch coast, and consists of 36 Vestas wind turbines of 3 MW each. The turbines are modelled in FarmFlow through their power and thrust curves (CERC, 2016).

For the OWEZ wind farm, nominal and robust AWC controllers have been designed. The robust design is performed by taking into account the same uncertain quantities as in Sect. 3.2 and 3.3, i.e. wind direction, wind speed, yaw error and turbulence intensity. For this wind farm, wind field time series of duration 6 h, sampled at 10 s, have been generated using
the approach in Sect. 2.1. The spacial resolution of the wind field is 200 m. On the average, the wind direction is 230°(South-West), the wind velocity is 8 m$s^{-1}$, and turbulence intensity is 7%. For the AWC's dynamic adaptation algorithm, the optimized





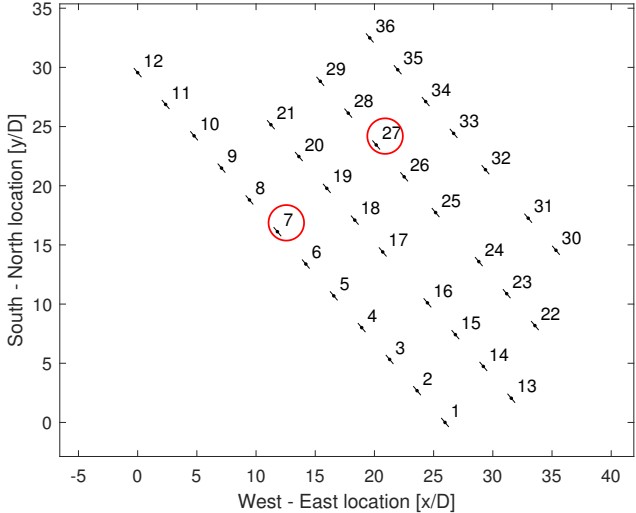

**Figure 13.** Layout of the OWEZ wind farm with the rotors oriented towards the average wind direction in the simulation, and with two selected for scoping wind turbines encircled.

parameters from Sect. 4 have been used. The same adaptation parameters have been used with both nominal and robust AWC to enable evaluation of the added value of performing robust AWC optimization over the conventional nominal design.

Three dynamic simulations have been carried out, one without AWC (serving as reference case), one with nominal AWC
and one with robust AWC. Each of these simulations took around 6 hours to complete on a single core. As an illustration, the local wind direction and the nacelle orientation for two selected wind turbines, those encircled in the layout plot in Fig. 13, are given in Fig. 14 for the three mentioned cases. Turbine T7 (left-hand side plot) is operating in free stream conditions for the simulated wind direction, while turbine T27 is in double wake situation and, hence, experiencing larger variations in the wind direction in accordance with the modelling described in Sect. 2.3. Because of that, the nacelle of T27 makes more often
excursions than that of T7 in the reference case without AWC (black lines in the figure), resulting in higher yaw duty. This becomes even more pronounced when looking at the yaw motion under nominal (blue curves) and robust AWC (red curves).

Finally, the results from the simulations have been evaluated in terms of the KPIs, defined in Sect. 4. The results are summarized in Table 4 for the three simulated cases. Besides the absolute values of the KPIs for the three scenarios, the table provides between brackets the relative increases with respect to the reference case without AWC. The following observations can be
made from the table:

- In terms of energy production, dynamic robust AWC improves significantly over the nominal AWC (2.05% vs 0.56% energy gain). Notice that the overall gains are lower than one might expect, which is to a large extend due to the somewhat irregular layout, especially in the lower right and upper left parts of the farm.

- The yaw travel, both average and worst-case, is significantly lower under robust AWC, which is partially due to the lower
yaw misalignment set-points under this strategy as compared to nominal AWC. Compared to the reference case without



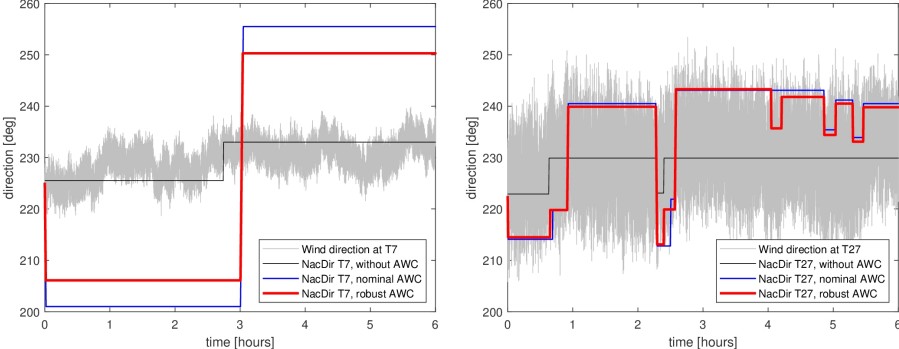

**Figure 14.** Wind direction and nacelle orientation for two selected wind turbines, $T7$ and $T27$, and three cases: without AWC, with dynamic nominal AWC and dynamic robust AWC. The sizes of the rotors are exaggerated by 50% for better readability of the figure.

**Table 4.** Summary of results from OWEZ simulations

| KPI | No AWC | dynamic nominal AWC (increase) | dynamic robust AWC (increase) |
|---|---|---|---|
| Energy production [MWh] | 166.19 | 167.13 (0.56%) | 169.59 (2.05%) |
| Average yaw travel [°] | 23.8 | 96.8 (×4.1) | 63.5 (×2.7) |
| Worst-case yaw travel [°] | 87.3 | 399.5 (×4.6) | 217.3 (×2.5) |
| Average nr. yaw events [-] | 3.5 | 7.7 (×2.2) | 5.3 (×1.5) |
| Worst-case nr. yaw events [-] | 13 | 30 (×2.3) | 14 (×1.1) |

AWC, the yaw travel under dynamic robust AWC is higher (factor 2.5-2.7), which is primarily due to the transitions between positive and negative misalignment. Notice that the reported values are only representative for the simulated wind conditions (with, on the average, wind direction aligned with turbine rows), and will be significantly lower on annual basis.

– The number of start/stop yaw events with the robust controller are also lower than with the nominal one. The increase with respect to the reference case is relatively low in this case (1.1-1.5), which on annual basis is expected to be even lower.

## 6   Conclusions

This paper considers the design of dynamic robust AWC that aims at optimizing the balance between the yaw duty and the

power gain in realistic conditions, i.e. in the presence of wind resource variability and measurement and model uncertainty. The starting point was an uncertainty quantification analysis performed for the following variables: wind speed, wind direction, yaw error, turbulence intensity, wind shear, air density, power curve, thrust curve, power loss coefficient due to yawed error. This





analysis indicated that the wind direction, yaw error, turbulence intensity and the wind velocity are the most important quantities to include in a robust AWC optimization. To this end, the variabilities and uncertainties are modelled as stochastic processes with corresponding PDFs, and the yaw misalignment set-point optimization is addressed as a stochastic programming problem through discretization of the probability distributions to arrive at a finite number of scenarios. A stationary analysis based on stochastic averaging using the PDFs of the uncertain parameters indicated that the robust AWC design slightly outperforms the nominal one in terms of power gain.

Subsequently, the design of the dynamic adaptation algorithm is considered, for the purpose of which the originally stationary FarmFlow wake model has been extended to enable dynamic simulations, including wake meandering and dynamic yaw control. The dynamic simulation model is fed by a realistic wind field including temporal and spacial inflow variations, with variations ranging from the micro-scale to meso-scale. Additional wind measurement noise is added for turbines operating in wake conditions, dependent on the local turbulence intensity, to model the increased yaw activity of downstream turbines, as observed in real-life measurements. Next, an AWC dynamic adaptation algorithm is considered, consisting of a low-pass filter, a hysteresis, and sample and hold mechanism. The parameters of these three building blocks have been optimized through a range of dynamic simulations with a five turbine array. It is shown that large-size hysteresis ($\pm 4°$) in combination with a low-pass filter with 60 s time constant and 10 s AWC sample time achieve the best trade-off between power and yaw duty. With these parameters, a reduction in the average number of yaw start/stop events of almost 50% and a power gain of 12% with respect to the reference scenario without AWC was achieved for the considered simulation. The yaw travel increased on the average by 13%, but its worst-case value over the turbines decreased by 5%.

Finally, the dynamic robust AWC approach is evaluated on a full-scale offshore wind farm model, for which both nominal and robust AWC controllers have been designed. The same (optimized) adaptation parameters have been used with both nominal and robust AWC to enable evaluation of the added value of performing robust AWC optimization. In the dynamic simulations performed, the robust AWC solution significantly improved over the nominal AWC one in terms of both power gain (2.05% vs 0.56% increase) and yaw duty (around 30-50% lower). Compared to the case without AWC, dynamic robust AWC increases the yaw duty somewhat, especially in terms of yaw travel (factor 2.5-2.7 higher). However, in terms of number of start/stop events, which is probably a more relevant indicator for the yaw loading, the increase is less significant (10-50%). It is important to note that the reported values are only representative for the simulated wind conditions (with, on the average, wind direction aligned with turbine rows), and is expected to be lower on annual basis.

The initial results in this paper underline the importance of optimizing AWC with respect to the real-life operating conditions, including wind resource variations and uncertainty in the measurements and modelling. Some topics that require further investigation in the future are:

- performance analysis on annual basis: simulations need to be performed for the whole range of possible wind conditions are required to assess the annual impact of dynamic robust AWC on the power production and yaw duty.

- deterministic (measurement) uncertainties: the current study is based on a stochastic representation of variabilities and uncertainties. For some quantities, e.g. measurement bias or wake model parameter uncertainties, a deterministic setting





might be more suitable, in which the optimization ensures best performance in the worst case uncertainty scenario. As wind resource variability will remain of stochastic nature, the robust design methodology should account for both stochastic and deterministic uncertainties.

– joint PDF: in the present work, all considered stochastic quantities were assumed independent. In reality, there is dependency between some variables, e.g. turbulence intensity depends on the wind velocity. To capture such dependencies properly, joint PDFs need to be used.

    – yaw set-points driven by consensus wind direction: to further reduce uncertainty in the wind direction measurements and, ultimately, the yaw duty, it might be beneficial to feed the AWC algorithm with a "consensus" wind direction signal
constructed using measurements from several nearby turbines. An example for such an approach is the centralized yaw control method, originally developed in Bossanyi (2019) to reduce yaw duty on wind turbine level, which might be beneficial when used in AWC as well.

    – include global blockage effects in wake modelling: these effect, currently under investigation, are expected to play an important role in improving the modelling accuracy for large wind farms.

– LiDAR-based feedforward AWC: using forward-looking LiDARs at upstream wind turbines, in combination with feedforward AWC control, is expected to be quite beneficial due to the very slow dynamics of the yaw system.

    – wind direction dependent measurement offset: recent research (Raach et al., 2019, Sect. 8) indicates that when a turbine stands in a partial wake situation, its wind direction measurement is biased by up to a few degrees. Future research is required to properly model this phenomenon, analyse its impact on the performance of AWC and develop a compensating
scheme.

*Code availability.* The simulations have been performed with FarmFlow, a proprietary tool developed by TNO.

*Author contributions.* EB developed the dynamic FarmFlow model, wrote Sect. 2.2, and reviewed the manuscript. The rest of the work was carried out by SK.

*Competing interests.* No competing interests.

*Acknowledgements.* This work is carried out in the framework of the project Dynamic Robust Wind Farm Control (DySCon), which was partially funded by the *TKI Wind op Zee PPS toeslag* program of the Dutch Ministry of Economic Affairs.





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
