# Peer review of "Dynamic robust active wake control"

_Wind Energy Science, 2021_

## Referee Comment (RC1)

**Major comments:**
1. The most significant improvement for this manuscript would be an increased clarify and structure, in my eyes. Often, sections are very long and one loses track of the purpose of a section. I would very much like to see the text restructured into more subsections and paragraphs. For example:
   a) Introduction: separate subsections for modern challenges in large-scale implementation.
   b) Introduction: too, much information on the dynamic FarmFlow part. I think you can remove lines 103-110: from "the dynamic simulation..." until "...in real-life measurements." without losing any valuable information in the introduction.
   c) Section 2.3, page 8: why is the derivation of wind direction variability part of the "yaw model"? This, to me, should be part of the inflow/wind field model.
   d) Table 1: this is a table related to validation of the model choices. This seems somewhat out of place to me, since you are still explaining the fundamentals of the model.
   e) Section 2.4: why is this part of Section 2: "wind farm model"? Typically, the wake/wind farm controller is not considered to be part of the wind farm model, especially in this entire context. Perhaps instead this should become part of Section 3 and Section 3 should become "AWC design"
   f) Section 3.1: find a way to clearly separate each factor of uncertainty/parameter. Perhaps a bullet point list or subsections/paragraphs.
   g) Line 437: can start a new subsection *(see next comment)
   h) Latter half of Section 4 vs. Section 5. One shows a basic case study for 3 turbines, and the other shows a more realistic case study with OWEZ. To me, it would make sense to put them both in Section 5 and separate them into two subsections: one for verification/simple study case for understanding, and then one for a more realistic evaluation.
   i) I read that FarmFlow has been extended to include wake and yaw control dynamics. It also now accepts dynamic wind fields to drive the simulation, including temporal and spatial variations. These are great developments. I would very much appreciate any kind of validation of these new functions. However, with the paper already being as long as it is, perhaps it would be better to present the dynamic FarmFlow plus validation in a separate publication. This would also increase clarity in the current manuscript.

2. Please motivate certain statements with the right literature and avoid speculation.
   a) In the introduction, I read that the potential AEP gain with AWC is several percents. This seems very high and currently not too realistic based on the recent expert elicitation and field experiments that exist in the literature. Actually, the papers cited with this statement are simulation studies and are better replaced with the Wingerden et al. expert elicitation and field experiments from Howland, Fleming, Simley, Duc and Doekemeijer. This relates to the minor comment on citing literature.
   b) Line 33, where the author assumes that the accumulated loads over the whole lifetime of a wind turbine decreases with wake steering, rather than increases, because Siemens-Gamesa is selling a wake steering solution. This reasoning seems flawed to me. We do not fully know under what conditions Siemens-Gamesa is doing wake

steering, if they require additional equipment, whether and which loads increase and decrease, when they do, and whether this relates to fatigue or ultimate loads. There is too little information to make any conclusions based on the fact that Siemens-Gamesa is selling wake steering, besides perhaps that it has caught the interest of this OEM.

c) Line 134-135, it is stated that wind farm simulations require time scales of tens of seconds. How about wake meandering or finer flow effects? How about large-eddy simulations? Add citations or at least defend this statement. Similarly, motivate choices of spatial resolution and sample time of the inflow.

d) Line 302: motivate natural frequency of meandering

e) Would be nice to clearly define the novel contributions in this article vs. what was done in previous work. FarmFlow already existed, but has been made dynamic: that is new, no? Uncertainty quantification is novel, at least for that exhaustive of a parameter set. Robust AWC and hysteresis already existed in literature, right?

**Minor comments:**

1. The abstract contains the general outline of the paper but misses the actual contributions and results. It currently does not suffice as a standalone summary of the paper. Please include the core findings, qualitatively but also quantitatively. For example, depict the parameters that were found to be the most important from the sensivity analysis, depict the potential AEP gain in percent, and so on.

2. Generally, and especially when citing literature, you should clarify the test environment used in that publication. The differences between a FLORIS-based simulation study, a SOWFA-based simulation study, a field experiment or a wind tunnel experiment are very significant.

3. For literature review: similar work is from M. Sinner et al., 2021, but this only appeared in April 2021. I can understand that the authors had already finished this publication mostly by then. You could consider including it in a revision.

   "Power increases using wind direction spatial
     filtering for wind farm control: Evaluation using
     FLORIS, modified for dynamic settings", Sinner et al., 2021, JRSE

4. Line 96: "This analysis ... the wind velocity." This is a conclusion and should not be part of the introduction. Rather, the introduction should be limited to what topics will be addressed in the article. The same goes for the sentence starting at

5. line 99: "A stationary analysis ... of power gain." Nice, but should go to conclusion.
   Line 182: "wake generated by a wind turbine is propagated downstream based on the local wind direction variations in its way", I do not understand this.

6. Line 183: "because the traveling time ... current time window." I do not understand this.

7. Line 202: "written in an output file", this seems inefficient. Can this not directly be exchanged through memory or over a network protocol?

8. Line 260: On what signal does the LP filter work?

9. Line 314: You mention that the PDF for turbulence intensity is based on historical data. Does your definition of TI (i.e., being the standard deviation in streamwise direction, match up with the definition in the data? I can imagine that the historical data considers the TI to include both streamwise and cross-stream turbulence.

10. Line 349: What optimization algorithm is used? How confident are you that the solution has converged? What are the bounds, e.g., have you limited the minimum and maximum yaw angles?
11. Table 3: the wind speed range seems so high, while in reality you could feed in the wind speed measurements into the LUT, perhaps with an uncertainty bound but definitely smaller than an uncertainty of 8 m/s. How do you defend this decision? Also, how do these findings line up with your earlier work stating that wind speed can be ignored in yaw optimizations?
12. Line 381: I would have expected the yaw-induced power loss coefficient to have a larger effect on the optimal yaw angles, since it directly impacts the energy lost by yawing an upstream wind turbine. Can you reason why this is not the case in this study?
13. Figure 5: yaw angles of -40 deg and + 45 deg seem excessive. Can you explain your choice for allowing yaw angles to go all the way to these values? Since we would never optimize the yaw angles until those limits in practice, this may skew the sensitivity analysis somewhat, no? Perhaps certain parameters are important at high misalignment angles, but really are not that important in the range we expect to yaw the turbines to.
14. Figures 5 & 6: please add legends
15. Line 414: should it read 'arg max' instead of 'arg min'?
16. Line 434: "...only decent directions...", what are "decent directions"?
17. Figure 8: neither line is particularly smooth. Does this suggest that the optimization has not converged?
18. Figure 9: "robust AWC" and "nominal AWC (with uncertainty)" are not the same thing, yet it is hard to distinguish them in their definitions. Can you clarify?
19. Figure 11: 11% Energy gain is very substantial and not particularly realistic for AEP. Maybe repeat that this is for particular 3-turbine case. Also, these figures are hard to see. I would suggest turning them into top-view (2D) contour plots instead. Same goes for Figure 12.
20. Line 555: Just to clarify, so dynamic FarmFlow runs 1:1 (6 hours of simulation means 6 hours of computing in real time on a single core)? If so, it may be worth evaluating the potential for a full year of operation (~9,000 CPU hours).
21. Line 567: "Notice that the overall gains are lower than one might expect", what would be a reasonable number to expect? 0.5-2% energy gain is still significant if you ask me.

**Technical comments:**
1. Variables should be italic, units should not.
2. Line 4: "by up to a few percentage points." Why percentage points and not percent?
3. Line 19: "possible power gains of up to a few percent on annual basis". Can you motivate this further, maybe add citations? To me, it seems that it is more towards a single percent, especially when looking at the most recent field experiments.
4. Line 20: The second challenge is presented as being mainly due to wake models being of static nature.
5. Figure 1: it says "yaw systems". Should it be "yaw system" since its for a single turbine?
6. Figure 1: The Robust AWC LUT seems only a function of wind direction and wind speed. Does this mean local WS/WD?
7. Line 149: "frequencies above 10e-3 Hz", should this be "above 10e-3 Hz"?
8. Line 279: "In this work ... robust design setting." I understand what you mean, but perhaps reformulate it in a clearer way. For example, differentiate between variables included as

uncertainties in the optimization process and variables that are used for the real-time interpolation of setpoints.

---

## Editor Comment (EC1)

In general this is a well-written research report with data that confirms existing results/expectations. The authors set-up their own complete toolchain to do the full analysis were some of the elements have been proposed by others or published in earlier work. This is also my major concern about this paper. The authors have to explain many steps and alter their own in-house design codes while some of the methods (maybe slightly different) have been published elsewhere or are available in an open source setting. Now the authors need a lot of tool development and explanation to reach their final conclusion. I believe that the paper would have been much stronger if connections were made to work by others (e.g. uncertainty quantification framework). Still, I believe that the final conclusion is important for the wind farm control research community.

Here some remarks:

Pg 1. "considered as the most potential technology" please add a citation

Pg 3. "is quite unrealistic" also a rather bold statement

It is "spatial" instead of "special"?

Section 2, at the beginning they talk about different sample rates (yaw vs flow) and there they question arises if you can do that. Later in the article they explain that the flow has two time scales (fast and slow).

Fig. 1. is really nice

Pg. 7. "because" …"because" (two sentences in a row)

Pg. 7. How the steps are explained it is just not clear. It is also not relevant that you are working with DLL's

Section 3, this should really be connected to ongoing efforts/frameworks

Pg 22. Figure 11 is not clear, Figure 12 is better (color) but still hard to read

Conclusion: is a conclusion of a research report and should be shortened. It should answer the research question (is there a research question?).

---

## Author Comment (AC1)

**Response to Reviewer RC1**

Dear Bart, thank you so much for the kind words, the thorough review of our manuscript and the numerous suggestions for improvements. Below, we have listed your comments and have provided our response after each one. The changes made to the manuscript are indicated by the boxed texts.

**Major comments:**

- The most significant improvement for this manuscript would be an increased clarify and structure, in my eyes. Often, sections are very long and one loses track of the purpose of a section. I would very much like to see the text restructured into more subsections and paragraphs. For example:
  - a. Introduction: separate subsections for modern challenges in large-scale implementation.
     **Response**: Thanks for the suggestion. We have added the following headings in the Introduction section: AWC implementation challenges, State of the art, Contribution of this work, Structure of the paper
  - b. Introduction: too, much information on the dynamic FarmFlow part. I think you can remove lines 103-110: from "the dynamic simulation..." until "...in real-life measurements." without losing any valuable information in the introduction.
     **Response**: We have removed these lines from the manuscript.
  - c. Section 2.3, page 8: why is the derivation of wind direction variability part of the "yaw model"? This, to me, should be part of the inflow/wind field model. **Response**: A good point! The organization of Section 2 was not clear enough, and we have now made an attempt to clarify the structure. To this end, Figure 1 has been updated to clearly indicate the four main components of the simulation model: wind field generator, dynamic wake model (dynamic FarmFlow), wind turbines' yaw model, and dynamic robust AWC. The yaw models and the dynamic robust AWC are implemented in a DLL, which is called by the dynamic FarmFlow code at each simulation step. Therefore, the yaw model is not part of the FarmFlow model (as one might probably expect), but is implemented separately in the DLL. Same holds for the added noise to the wind direction signals, coming from FarmFlow, which is included to model the effect of the increased turbulence in the wake on the measured wind direction. To clarify the structure of Section 2, next to the clarifications made in Figure 1, we have modified the name of Section 2 to "Simulation model description" and have updated the text in the beginning of Section 2 as follows:

"The wind farm simulation model consists of a stochastic wind field generator, dynamic wake model (dynamic FarmFlow), wind turbines' yaw model, and dynamic robust AWC. A block scheme of the simulation model is given in Fig. 1, in which the four mentioned main components have been clearly indicated. The yaw models and the dynamic robust AWC are implemented in a dynamic link library (DLL), which is called by the dynamic FarmFlow code at each simulation step...

...The main components are explained separately in more detail in the remainder of this section."

d. Table 1: this is a table related to validation of the model choices. This seems somewhat out of place to me, since you are still explaining the fundamentals of the model.

**Response**: The table does seem somewhat misplaced here, indeed, but we could not find any better place for it. It is included to support the idea of adding turbulencedependent noise term in the wind direction measurements entering the yaw model, resulting in increased yawing of downstream turbines. We could remove the table and the related text, but we do believe it adds value to the discussion. Please, feel free to make a specific advice as to where we could better place the table.

e. Section 2.4: why is this part of Section 2: "wind farm model"? Typically, the wake/wind farm controller is not considered to be part of the wind farm model, especially in this entire context. Perhaps instead this should become part of Section 3 and Section 3 should become "AWC design" **Response**: We believe this comment is much related to comment (c) above, i.e. the lack of clarity of the structure of Section 2. As explained in our response to comment (c) above, the structure has been clarified in the revised manuscript. Section 2 describes the complete simulation model which consists of four main components, as depicted in Fig.1. These are wind field generator, dynamic wake model (dynamic FarmFlow), wind turbines' yaw model, and dynamic robust AWC. Each component of the simulation model is then described in a separate subsection. Section 3 "Robust AWC optimization" is on the optimization of the parameters of the robust AWC (the LuT in Fig. 1), which Section 4 -on the optimization of the dynamic part of the AWC algorithm (LP filter, hysteresis and sampling). To clarify this further in the manuscript, we have included the following text in the beginning of Section 2.4: "This section describes the structure of the dynamic robust AWC algorithm,

"This section describes the structure of the dynamic robust AWC algorithm, represented by the shaded area at the bottom of Fig. 1. The optimization of the parameters of the underlying blocks of this algorithm is topic of Sect. 3 (Robust AWC optimization) and 4 (Dynamic adaptation algorithm optimization)."

- f. Section 3.1: find a way to clearly separate each factor of uncertainty/parameter. Perhaps a bullet point list or subsections/paragraphs.
   **Response**: we have improved the structure to Section 3.1 in the revised manuscript by using a list.
- g. Line 437: can start a new subsection \*(see next comment)
   Response: see response to next comment.
- h. Latter half of Section 4 vs. Section 5. One shows a basic case study for 3 turbines, and the other shows a more realistic case study with OWEZ. To me, it would make sense to put them both in Section 5 and separate them into two subsections: one for verification/simple study case for understanding, and then one for a more realistic evaluation.

**Response**: Good idea! We moved the simplified example with 5 turbines from Section 4 to the beginning of Section 5, and making Section 5.1 out of it. The realistic case study with OWEZ became Section 5.2. We added the following text to the beginning of Section 5 to classify this: "This section presents the results from two case studies. The first one represents a example of robust AWC design performed for a simple farm consisting of a few turbines in a row. The second one represents a realistic case study with dynamic robust AWC applied to an offshore wind farm."

i. I read that FarmFlow has been extended to include wake and yaw control dynamics. It also now accepts dynamic wind fields to drive the simulation, including temporal and spatial variations. These are great developments. I would very much appreciate any kind of validation of these new functions. However, with the paper already being as long as it is, perhaps it would be better to present the dynamic FarmFlow plus validation in a separate publication. This would also increase clarity in the current manuscript.

**Response**: Very valid point. The dynamic part of the FarmFlow model has, unfortunately, not been properly validated yet, and we are currently looking for funding to support this very important work. Some sanity checks have, of course, been done to ensure the output makes sense, but a more rigorous validation is still needed. Once this is done, we will of course consider publication.

- 2. Please motivate certain statements with the right literature and avoid speculation
  - a. In the introduction, I read that the potential AEP gain with AWC is several percents. This seems very high and currently not too realistic based on the recent expert elicitation and field experiments that exist in the literature. Actually, the papers cited with this statement are simulation studies and are better replaced with the Wingerden et al. expert elicitation and field experiments from Howland, Fleming, Simley, Duc and Doekemeijer. This relates to the minor comment on citing literature.

**Response**: Several percent AEP increase is not stated anywhere in the manuscript. Instead, both in the Abstract and in the Introduction a *possible, or potential, gain of up to a few percent* is mentioned. In my opinion, this is not exaggerating the current knowledge. Indeed, this statement is only backed up by numerical studies, but the field experiments are currently only very limited and provide no basis for estimation of the AEP gains achievable in the future. Furthermore, the paper on expert elicitations does not cover the question of what is the expected AEP increase, but rather the following one: "How much of an increase in energy production is needed to justify implementation?". In our opinion, due to lack of other evidence, we believe it is better to stick to our initial citations.

b. Line 33, where the author assumes that the accumulated loads over the whole lifetime of a wind turbine decreases with wake steering, rather than increases, because Siemens-Gamesa is selling a wake steering solution. This reasoning seems flawed to me. We do not fully know under what conditions Siemens-Gamesa is doing wake steering, if they require additional equipment, whether and which loads increase and decrease, when they do, and whether this relates to fatigue or ultimate loads. There is too little information to make any conclusions based on the fact that Siemens-Gamesa is selling wake steering, besides perhaps that it has caught the interest of this OEM.

**Response**: It is not true that we assume that the lifetime fatigue loads decrease with wake steering because SGRE is selling such a solution. Instead, the manuscript

clearly states: "A more detailed study involving a utility scale wind farm was presented in Kanev et al. (2020), where the impact of wake steering AWC on the structural loads of the turbines during their complete lifetime has been investigated using the so-called loads lookup table (LUT) approach (Reyes et al. 2020). The results demonstrate that, even though by itself yaw misalignment does increase the structural loads of some turbines in specific wind conditions, the wake-induced loading is decreased even more, so that the accumulated loads over the whole lifetime of each wind turbine generally remain lower than without AWC.". Nevertheless, we have removed the following sentence in the revised manuscript to avoid any misinterpretations:

"This conclusion is implicitly confirmed by the fact that the industry starts to develop this technology into commercial products (Siemens Gamesa Renewable Energy, 2019)."

c. Line 134-135, it is stated that wind farm simulations require time scales of tens of seconds. How about wake meandering or finer flow effects? How about large-eddy simulations? Add citations or at least defend this statement. Similarly, motivate choices of spatial resolution and sample time of the inflow.
 Response: The wind farm modelling required for this study needs only to include

effects necessary for modeling the power production of the wind turbines, the wake meandering, and the yaw dynamics. Higher frequencies (>1Hz) are not correlated across the wind farm and represent local turbulence variations. These are important for the turbine loads, of course, but loads modeling falls outside the scope of the model. Higher frequencies are, however, relevant for the proper modeling of the nacelle yaw dynamics, and are therefore included into the model (but uncorrelated in space). The modelling approach, followed in this work, is actually quite well aligned with earlier work of others: Bossanyi (2018) and Smiley (2020). We have now indicated this in the revised manuscript through the following addition at the end of the first paragraph of Section 2.1:

"Notice that this approach is quite similar to those followed by Bossanyi (2018) and Simley et al. (2020), where the authors also split the wind field spectrum into lowfrequency (for the wake dynamics) and high-frequency (for the turbine yaw dynamics). In Bossanyi (2018), the same spacial resolution is used for the lowfrequency wind field as well."

Regarding the sampling time, the text states already that it is

".... roughly equal the time it takes air flow to cover a distance of 2D."

Given the fact that 2D is the special resolution, we believe this makes sense.

d. Line 302: motivate natural frequency of meandering

**Response**: We could not find the text (around line 302) to which this comment relates. However, for the good order, we added a relevant citation in the first paragraph of Section 2.1 in the revised manuscript, regarding the wake meandering modeling on which our choice of 2D for the special resolution is based:

"The chosen spacial resolution of the wind fields is around two rotor diameters (2D), which is in accordance with the wake meandering modelling in Larsen et al. (2008)."

e. Would be nice to clearly define the novel contributions in this article vs. what was done in previous work. FarmFlow already existed, but has been made dynamic: that is new, no? Uncertainty quantification is novel, at least for that exhaustive of a parameter set. Robust AWC and hysteresis already existed in literature, right? **Response**: Good point. As already mentioned in our response to 1 a), we added a heading "Contribution of this work" in the introduction. We have now also added a summary of the main contributions of the paper for more clarity:

"In summary, the main contributions in this work are as follows:

- Development of dynamic wake model, based on the originally static FarmFlow tool, suitable for design and evaluation of AWC solutions.
- Exhaustive uncertainty quantification analysis, pinpointing the most important uncertainty contributors that need to be considered in a robust AWC design setting.
- Optimization of the parameters of a dynamic AWC algorithm using a wide range of dynamic simulations, with the purpose of maximizing the power gain and minimizing the yaw duty.
- Design and evaluation of a dynamic robust AWC algorithm for a realistic case study with a full scale wind farm."

**Minor comments**

 The abstract contains the general outline of the paper but misses the actual contributions and results. It currently does not suffice as a standalone summary of the paper. Please include the core findings, qualitatively but also quantitatively. For example, depict the parameters that were found to be the most important from the sensivity analysis, depict the potential AEP gain in percent, and so on.

**Response**: We have extended the abstract as suggested, by including the following text:

"To this end, an uncertainty quantification analysis has first been performed for a range of variables (wind speed, wind direction, yaw error, turbulence intensity, wind shear, air density, power curve, thrust curve, power loss coefficient due to yawed error), which indicated the wind direction, yaw error, turbulence intensity and the wind velocity as the highest uncertainty contributors. Robust AWC has next been synthesized by including stochastic uncertainties in these parameters. A stationary analysis through stochastic averaging indicated that the robust AWC design only slightly outperforms the nominal one in terms of power gain. For the dynamic design and analysis, the originally stationary FarmFlow wake model has been extended to enable dynamic simulations, including wake dynamics and a dynamic yaw control model. By selecting a certain dynamic adaptation algorithm structure (a low-pass filter, hysteresis, and sample and hold mechanism), a wide range of dynamic simulations has been performed to optimize its parameters for achieving the best balance between power gain and yaw duty. Dynamic simulations for a realistic case study with a full-scale wind farm indicated that the developed dynamic robust AWC results in a large reduction of the yaw duty (30-50% lower) while at the same time improving the overall power gain (2.05% vs. 0.56%), as compared to the conventional nominal AWC."

2. Generally, and especially when citing literature, you should clarify the test environment used in that publication. The differences between a FLORIS-based simulation study, a SOWFA-based simulation study, a field experiment or a wind tunnel experiment are very significant.

**Response**: Of course, and we believe we have done that in many places in the original manuscript, such as on Line 55 "...in recent field studies with wake redirection (Fleming et.al., 2020, 2019)", Lines 67-69 "Above-mentioned studies on robust AWC were all performed using a simplified control-oriented wake model, namely the FLOw Redirection and Induction in Steady State (FLORIS) model - an understandable choice given the computational requirements for robust optimization.", Lines 70-71 "...utilizing a different steady-state wake model, called the lifting line model.", Line 83 "In Smiley et.al (2020), for instance, dynamic simulations have been performed using the stationary FLORIS wake model", etc. It is unclear to us what the point of this comment is.

3. For literature review: similar work is from M. Sinner et al., 2021, but this only appeared in April 2021. I can understand that the authors had already finished this publication mostly by then. You could consider including it in a revision.

"Power increases using wind direction spatial filtering for wind farm control: Evaluation using FLORIS, modified for dynamic settings", Sinner et al., 2021, JRSE

**Response**: Thanks for pointing us out to this relevant recent publication, we have of course included a citation in the revised manuscript through the following text in the introduction:

"Combining these techniques with wake steering was considered recently in the work of Sinner et al. (2021) using a modified FLORIS model."

4. Line 96: "This analysis ... the wind velocity." This is a conclusion and should not be part of the introduction. Rather, the introduction should be limited to what topics will be addressed in the article. The same goes for the sentence starting at

line 99: "A stationary analysis ... of power gain." Nice, but should go to conclusion. **Response**: Agreed. The sentence on Line 96 of the original manuscript has been removed, and the next sentence modified to:

"Based on the results from this analysis, robust yaw misalignment set-points have been optimized with respect to the most significant uncertainty sources, modelled as independent stochastic processes with selected PDFs."

The sentence commencing at Line 99 has been modified to:

"A stationary analysis based on stochastic averaging has been carried out to evaluate the performance of the robust AWC design as compared to the nominal one in terms of power gain."

Line 182: "wake generated by a wind turbine is propagated downstream based on the local wind direction variations in its way", I do not understand this.
 Response: The sentence has been rephrased as follows:

"To this end, the wake generated by a wind turbine is propagated downstream in such a way that it follows on its way the local wind direction variations in the wind field. This way, both time delay and meandering effects are modelled."

6. Line 183: "because the traveling time ... current time window." I do not understand this. **Response**: This sentence has been rephrased as follows:

"Since the travel time of a wake between two turbines takes longer than the simulation sample time, ..."

7. Line 202: "written in an output file", this seems inefficient. Can this not directly be

exchanged through memory or over a network protocol? **Response**: This certainly can, and might be implemented in future updates of the software tool.

8. Line 260: On what signal does the LP filter work?
 **Response**: The following clarification was added to the LP filter description:
 "As visualized in Fig. 1, the LP filter acts on the wind speed and wind direction signals."

9. Line 314: You mention that the PDF for turbulence intensity is based on historical data. Does your definition of TI (i.e., being the standard deviation in streamwise direction, match up with the definition in the data? I can imagine that the historical data considers the TI to include both streamwise and cross-stream turbulence.

**Response**: Our definition of TI in the wake model and in the wind field generator is, in fact, pretty standard, and as such are strictly speaking not exactly matching the definition TI measurements based on 10 minute statistical met mast data (mean value and standard deviation of the measured wind velocity). However, notice that these data is used to construct a rough, though realistic, statistical model of the turbulence intensity *variations*, which serves primarily as an example to based the robust analysis and design on. We do not believe that tinkering around the edges here will improve the value of the paper.

10. Line 349: What optimization algorithm is used? How confident are you that the solution has converged? What are the bounds, e.g., have you limited the minimum and maximum yaw angles?

**Response**: The following text has been added in Section 3.2 to explain the optimization algorithm used in this study:

"To solve the underlying optimization problems, a tailor made algorithm has been used that requires a minimum number of function evaluations (farm simulations) to converge. The algorithm is similar to the conventional bisection method, but generalized to multivariate objective functions. By confining the optimization variable to lie within an initial ndimensional box, the gradient of the objective function is evaluated at the centre point at each iteration and the box is reduced in size by keeping only that part that is oriented opposite to the gradient. While this algorithm has no theoretical guarantees to converge to an optimum solution for general nonlinear functions, has been successfully used for many years by the authors and works pretty well for the application at hand, its low calculation effort being its main advantage over alternative algorithms. This allows it to be used in combination with relatively complex wake models such as FarmFlow. To reduce computation time even more, the number of optimization variables is limited to the yaw setpoints of the two most upstream turbines in each row of turbines oriented downstream. The yaw set-points for the remaining turbines in the row are linearly decreased between the second turbine and the last one, which has zero yaw misalignment set-point. No limitation has been applied to the yaw set-points in this section."

To clarify that this algorithm is used in all optimizations throughout the paper, the following like is added at the end of Section 3.3:

"All optimization problems are solved by using the algorithm, described in Sect. 3.2. The yaw misalignment angles have been limited to  $\pm 30$  degrees."

11. Table 3: the wind speed range seems so high, while in reality you could feed in the wind speed measurements into the LUT, perhaps with an uncertainty bound but definitely smaller than an uncertainty of 8 m/s. How do you defend this decision? Also, how do these findings line up with your earlier work stating that wind speed can be ignored in yaw optimizations? **Response**: In fact, the idea is to not use the wind speeds as an argument for the LUT (other than for switching AWC on and off), but rather to have the LUT robust with respect to the whole range of wind speed variations. This is mentioned in the first paragraph of Section 3, as well as in the second bullet point in Section 3.1, where the modeling of the wind speed uncertainty is described. Nevertheless, to better clarify this, the following line is added to the mentioned bullet point:

"Instead, the LUT will be designed to be robust to wind speed variations."

- 12. Line 381: I would have expected the yaw-induced power loss coefficient to have a larger effect on the optimal yaw angles, since it directly impacts the energy lost by yawing an upstream wind turbine. Can you reason why this is not the case in this study? Response: You are absolutely right. Large variations in the yaw-induced power loss coefficient result in significant variations in the optimal yaw set-points. Take, for instance the result for Case 2 in Figure 5, depicted by the middle (green) bar plot. The optimal yaw misalignment of the first turbine is around 22° for yaw-induced power loss coefficient of 2.3, and 35° for a coefficient of 1.3. Computing the corresponding power losses,  $\cos(\beta)^{\alpha}$ , one gets 0.84 for  $\alpha$ =2.3 and  $\beta$ =22°, and 0.77 for a  $\alpha$ =1.3 and  $\beta$ =35°. The larger power loss at the first turbine in the later case is then compensated by the corresponding higher yaw misalignment, leading to increased power production downstream. So the results make perfect sense. Including the assumed distribution in Figure 4, however, narrows down the most probable range of variations of the yaw-induced power loss coefficient to below 1.7, leading to a quite small size of the green box in Figure 5. To summarize, the yaw-induced power loss coefficient does affect the optimal yaw set-points, but for the assumed uncertainty model (PDF) this impact remains limited and the parameter does not need to be considered in the robust optimization.
- 13. Figure 5: yaw angles of -40 deg and + 45 deg seem excessive. Can you explain your choice for allowing yaw angles to go all the way to these values? Since we would never optimize the yaw angles until those limits in practice, this may skew the sensitivity analysis somewhat, no? Perhaps certain parameters are important at high misalignment angles, but really are not that important in the range we expect to yaw the turbines to. **Response**: This is, in fact, quite a good point, and not touching upon it is indeed an omission. The optimizations in Section 3.2 have been consciously performed without applying limitations on the yaw misalignment, as the exact limitations that will apply in future applications are not exactly known and the results from the uncertainty quantification analysis should be generic. However, as limitations will apply in practice, we agree it is important to discuss their impact. We have added the following paragraph to Section 3.2 to discuss this important aspect:

"Notice that, due to the fact that no limitations have been imposed, the yaw misalignment set-points are getting quite high in some cases, raising to values of 40 degrees and even higher. Such a high yaw misalignments are currently considered unrealistic in a real-life application. Usually, they are limited to around 30 degrees in many research studies, or to even lower values in the first field tests with wake redirection (Flemming et al., 2017, 2019; Doekemeijer et al., 2021). It becomes clear from Fig. 5 that by imposing a limitation of ±30 degrees one would significantly limit the variation in the yaw misalignment set-points. Nevertheless, the main conclusions drawn above in terms of the most significant uncertainty contributors will still hold, with probably only the wind shear disappearing from this list."

- 14. Figures 5 & 6: please add legends: **Response**: Legends added to Figures 5 and 6!
- 15. Line 414: should it read 'arg max' instead of 'arg min'? **Response**: Of course it should. We have fixed this, as well as 4 other occurrences!
- 16. Line 434: "...only decent directions...", what are "decent directions"? **Response**: This sentence is not precent in the revised manuscript. Instead, as explained in the Response to question 10 above, the description of the optimization algorithm has been explained in detail in Section 3.2.
- 17. Figure 8: neither line is particularly smooth. Does this suggest that the optimization has not converged?

**Response**: This has to do with the nonlinearily of the objective function, the chosen optimization algorithm, and the termination criterion, indeed. It is, of course, possible to improve the optimizer to get smoother curves, but that would require increasing the calculation time significantly. We believe, however, that this is not really necessary since resulting power gain is not very sensitive on such relatively small variations of the yaw misalignment angles. This fact can be appreciated from Figure 12, noticing that the power gain barely changes between nominal and robust AWC. Smoothing of the yaw set-points can be easily done after the optimization, which does save a lot of calculation effort, especially when designing AWC for the complete spectrum of wind conditions in a full scale wind farm. Robust design does, however, deliver smoother yaw angles in general.

18. Figure 9: "robust AWC" and "nominal AWC (with uncertainty)" are not the same thing, yet it is hard to distinguish them in their definitions. Can you clarify? **Response**: We have added the following text at the end of Section 5.1 to clarify the different curves in the figure (now Figure 12 in the revised manuscript due to the change explained in the Response to Major comment 1h):

"For the sake of clarity, the difference between the red curve "nominal AWC (no uncert.)" and the black curve "nominal AWC (with uncert.)" in Fig. 12 is that the former one depicts the power gain evaluated just for the nominal values of the uncertainty parameters  $(p^nom)$ , while the later one represents the gain evaluated by including the whole uncertainty set *U* through the joint PDF  $D_{-d}(p)$ . In both cases, the yaw misalignment setpoints are the same, namely  $\gamma_{-nom}$ , optimized for the nominal values of the parameters  $p^nom$ , i.e. neglecting the uncertainty, as defined in Eq. (4). The blue curve "robust AWC" corresponds to the case when both the optimization of the yaw set-points and the evaluation of the power gain are performed by accounting for the uncertainty through  $D_{-d}(p)$ ." 19. Figure 11: 11% Energy gain is very substantial and not particularly realistic for AEP. Maybe repeat that this is for particular 3-turbine case. Also, these figures are hard to see. I would suggest turning them into top-view (2D) contour plots instead. Same goes for Figure 12. Response: The 11% gain is not for AEP but for the performed simulation conditions only, mean wind velocity of 8 m/s and wind direction varying around 270 degrees. 11% power gain is not at all unusual in such a scenario, as you would agree. We added a clarification about the simulation conditions in Section 4 as follows

"The average wind velocity is 8 ms-1, and the wind direction varies around 270° (see light grey line in Fig. 10)."

- Line 555: Just to clarify, so dynamic FarmFlow runs 1:1 (6 hours of simulation means 6 hours of computing in real time on a single core)? If so, it may be worth evaluating the potential for a full year of operation (~9,000 CPU hours).
   Response: Correct. A full year can easily be calculated on a computer cluster.
- 21. Line 567: "Notice that the overall gains are lower than one might expect", what would be a reasonable number to expect? 0.5-2% energy gain is still significant if you ask me. **Response**: Again, the power gains are not on annual basis (AEP), but for the simulated wind conditions only. You would agree that for the "more beneficial" wind directions the power gain is usually higher than that.

**Technical comments**

- Variables should be italic, units should not.
   Response: We have carefully gone through the text, including the tables, and changed all noticed occurrences of italic units to regular. The variables were properly typeset in the original manuscript already.
- Line 4: "by up to a few percentage points." Why percentage points and not percent? Response: "percentage points" changed to "percent".
- Line 19: "possible power gains of up to a few percent on annual basis". Can you motivate this further, maybe add citations? To me, it seems that it is more towards a single percent, especially when looking at the most recent field experiments.
   **Response**: This question is quite similar to that posed in Major comment 2a. Please, refer to our response to that comment.
- 4. Line 20: The second challenge is presented as being mainly due to wake models being of static nature.

**Response**: As stated in the text: "The second challenge is related to the uncertainty in the predictions for the expected annual energy production (AEP) increase, caused by the simplistic static approach that is currently used to optimize AWC on the one side, and the underlying uncertainties in the modelling used for that purpose on the other side."

5. Figure 1: it says "yaw systems". Should it be "yaw system" since its for a single turbine? **Response**: Yes, we changed it to "yaw system".

Figure 1: The Robust AWC LUT seems only a function of wind direction and wind speed. Does this mean local WS/WD?
 Response: The local wind conditions are used by the AWC algorithm. To clarify this in the text, the following line is added in the first paragraph of Section 2.4:
 "In the implementation, used in this work, the AWC algorithm receives the local wind speed.

"In the implementation, used in this work, the AWC algorithm receives the local wind speed and wind direction as measured at turbine level."

- Line 149: "frequencies above 10e-3 Hz", should this be "above 10e-3 Hz"?
   Response: I don't understand this comment. I believe the sentence is clear: "This Kaimal spectrum is used for frequencies above 10-3 Hz, i.e. time scales of 30 minutes and slower."
- 8. Line 279: "In this work ... robust design setting." I understand what you mean, but perhaps reformulate it in a clearer way. For example, differentiate between variables included as uncertainties in the optimization process and variables that are used for the real-time interpolation of setpoints.

**Response**: We have rewritten this text as follows to improve the clarity: "In this work, the yaw misalignment set-points in the LUT will essentially be a function of the wind direction only. The wind speed signal entering the block "Robust AWC (LUT)" in Fig. 1 is meant to indicate that AWC is only active in a certain range of below rated wind conditions (4-12 ms-1 used here)."

**Response to Reviewer RC2**

Dear reviewer, thank you so much for the kind words and the useful comments and suggestions for improvements. Below, we have listed your comments and have provided our response after each one. The changes made to the manuscript are indicated by the boxed texts.

**General Comments:**

- Section 2.1: Do I understand that wind directions variations occur in the range of 30min 24hr, and that faster frequencies are uncorrelated spatially? If we expect a wind turbine to yaw something like several times every 10 minutes does this match?
   **Response**: No, in fact both the micro-scale model (*f*>10-3 Hz) and the meso-scale model (*f*<10-3 Hz) include special correlation, as modelled in equation (1). Kindly note the sentence just before the equation stating "For the complex cross power spectrum between two points in space, *r* and *s*, the following expression is used for both the micro and the meso scale spectra ...".
- Could you provide a definition of stochastic programming in general and how it is used in this work?

**Response**: Definition of the stochastic programming problem, as well as explanation of how it is numerically solved in our work, are explained in detail in Section 3.3. Since the objective function cannot be evaluated in its continuous form, the joint PDF is first discretized and the resulting optimization problem is solved through the algorithm described in Section 3.2. This approach, based o discretization, has already been used by others in this application, and we have included citation to that work in the revised manuscript:

"To solve this problem numerically, the continuous PDFs are discretized as in Rott et al. (2018); Smiley et al. (2020)."

**Specific comments:**

• Page 2: "This conclusion is implicitly confirmed by the fact that the industry starts to develop this technology into commercial products (Siemens Gamesa Renewable Energy, 2019)." Could also be that the loads are higher but not importantly so?

**Response**: This is certainly true. The point we were trying to make is that the loads don't seem to be an obstacle for the implementation of AWC, but we have removed the following sentence in the revised manuscript to avoid any misinterpretations:

"This conclusion is implicitly confirmed by the fact that the industry starts to develop this technology into commercial products (Siemens Gamesa Renewable Energy, 2019)."

Page 3: "In a different work, the same author demonstrates that a centralized yaw control strategy, in which information from surrounding wind turbines is used in the yaw control algorithm, can lead to a drastic reduction in the yaw duty and increase the power capture at the same time (Bossanyi, 2019)." This could also be related to the concept of consensus control: Annoni, J., Bay, C., Johnson, K., Dall'Anese, E., Quon, E., Kemper, T., and Fleming, P.: Wind direction estimation using SCADA data with consensus-based optimization, Wind Energ. Sci., 4, 355–368, https://doi.org/10.5194/wes-4-355-2019, 2019.

**Response**: This is definitely a very relevant work, we have included a reference to it in the Introduction through the following text:

"Related to that is the work of Annoni et al. (2019) focused on constructing consensus wind direction estimates."

 This sentence: "This Kaimal spectrum is used for frequencies above 10–3 150 Hz, i.e. time scales of 30 minutes and slower". If the range is above a frequency, do you mean lower and not slower?

**Response**: Yes, slower is now changed to lower.

- Page 6: "The parameter c(αrs) is the decay factor". Decay of what?
   **Response**: We have added the following clarification regarding this parameter:
   "The parameter c(αrs) is the decay factor, a parameter characterizing the decay rate of the coherence function."
- Page 10: Recommend to explain figure 2 in more detail in the caption
   **Response**: We modified the caption of the figure as follows to make it more explanatory:
   "Illustration of the impact of wake on wind measurements and yaw motion of two commercial wind turbines, when the second turbine (T2) is in the wake of the first one (T1). The thin lines represent LP filtered wind direction measurements at the two turbines, while the think ones -- the nacelle position measurements."
- Page 13: Don't need to revise the paper, but wanted to note I think some recent papers might point to a distribution for yaw loss exponent centered somewhat higher, or even dependent on wind speed: Simley, E., Fleming, P., Girard, N., Alloin, L., Godefroy, E., and Duc, T.: Results from a Wake Steering Experiment at a Commercial Wind Plant: Investigating the Wind Speed Dependence of Wake Steering Performance, Wind Energ. Sci. Discuss. [preprint], https://doi.org/10.5194/wes-2021-61, in review, 2021.

**Response**: Of course, very relevant work. We have included a reference to it in the text as follows:

"Related to that is the work of Annoni et al. (2019) focused on constructing consensus wind direction estimates."

• Page 15. "Variations in the thrust curve and the yaw-induced power loss exponent have generally limited impact on the optimal yaw set-points, which suggests that they could be left out from the robust optimization." This is surprising, at least for the power curve exponent, it would seem that at some loss level it would start to have a strong impact,? **Response**: You are absolutely right. Large variations in the yaw-induced power loss coefficient result in significant variations in the optimal yaw set-points. Take, for instance the result for Case 2 in Figure 5, depicted by the middle (green) bar plot. The optimal yaw misalignment of the first turbine is around 22° for yaw-induced power loss coefficient of 2.3, and 35° for a coefficient of 1.3. Computing the corresponding power losses,  $\cos(\beta)^{\alpha}$ , one gets 0.84 for  $\alpha$ =2.3 and  $\beta$ =22°, and 0.77 for a  $\alpha$ =1.3 and  $\beta$ =35°. The larger power loss at the first turbine in the later case is then compensated by the corresponding higher yaw misalignment, leading to increased power production downstream. So the results make perfect sense. Including the assumed distribution in Figure 4, however, narrows down the most probable range of variations of the yaw-induced power loss coefficient to below 1.7, leading to a quite small size of the green box in Figure 5. To summarize, the yaw-induced power loss coefficient does affect the optimal yaw set-points, but for the assumed uncertainty model (PDF) this impact remains limited and the parameter does not need to be considered in the robust optimization.

- Page 19: "The yaw set-points for the remaining turbines in the row are linearly decreased between the second turbine and the last one, which has zero yaw misalignment set-point." This is a great idea! Is this novel to this paper or has it the concept been used elsewhere?
   Response: Well, it's not really new, this is an approach we use for years to reduce the computational time for the optimization. It is first mentioned in our work: Kanev, S.; Savenije, F. & Engels, W. Active wake control: an approach to optimize the lifetime operation of wind farms, Wind Energy, 2018, 21, 488-501
- Page 21: Metrics are really useful, the power gain per unit yaw travel increase is very interesting, is this also a novelty of this paper or something used in other papers or other contexts
   **Response**: Well, we haven't seen these metrics in other publications, but they are probably also not too difficult to invent when one tries to capture the wish to optimize the balance between yaw duty and power gain into a cost function.
- Page 23: Results for hysteresis are very promising. If 4 deg is both the highest value tested and the best overall, does it suggest 5 deg or more should be considered?
   **Response**: Yes, in fact it does. This should be considered in future studies. We have included the following comment in the revised manuscript:

"This result also suggests that even higher values for the hysteresis size be considered in future studies."

• Page 24: "Having significantly less start/stop events in the reference case than with AWC might first seem couter-intuitive, but does happen". Did this sentence mean to say the reverse (49% reduction)?

**Response**: You are completely right, thanks for noticing. We have corrected the sentence as follows:

"Having significantly less start/stop events with AWC than in the reference case might first seem counter-intuitive, but does happen."

Page 28: Recommend to cite paper mentioned above on consensus control
 **Response**: We added reference to the mentioned work as follows in the Conclusions:

"Examples such approaches for constructing consensus wind directions can be found in Annoni et al. (2019) and Bossanyi (2019), which are expected to be beneficial for AWC as well."

**Dynamic robust active wake control**

Stoyan Kanev1 and Edwin Bot1

1TNO Energy Transition, Wind Energy, Westerduinweg 3, 1755LE Petten, Netherlands **Correspondence:** Stoyan Kanev (stoyan.kanev@tno.nl)

**Abstract.** Active Wake Control (AWC) is a strategy for operating wind farms in a way to maximize the overall power production and/or reduce structural loading on the wind turbines. Many recent studies indicate that this technology, and more specifically the so-called wake redirection approach to AWC, have a significant potential for increasing the annual energy production (AEP) by up to a few percentage pointspercent. The current state-of-the-art approach is to optimize AWC for a range

- 5 of static wind conditions, which is expected to perform sub-optimally in real-life due to the continuous variations of the wind resource and the very slow yaw dynamics of the turbines. Recent work has addressed this variability in a robust design setting with the focus on maximizing the energy capture (robust AWC). This paper continues on this line of research, and develops a *dynamic* robust AWC strategy that aims to optimize the balance between maximum power production (requiring increased level of yawing) and minimum loads on the yaw drive (requiring limited yaw motion). It is shown with a
- 10 To this end, an uncertainty quantification analysis has first been performed for a range of variables (wind speed, wind direction, yaw error, turbulence intensity, wind shear, air density, power curve, thrust curve, power loss coefficient due to yawed error), which indicated the wind direction, yaw error, turbulence intensity and the wind velocity as the highest uncertainty contributors. Robust AWC has next been synthesized by including stochastic uncertainties in these parameters. A stationary analysis through stochastic averaging indicated that the robust AWC design only slightly outperforms the nominal one in terms
- 15 of power gain. For the dynamic design and analysis, the originally stationary FarmFlow wake model has been extended to enable dynamic simulations, including wake dynamics and a dynamic yaw control model. By selecting a certain dynamic adaptation algorithm structure (a low-pass filter, hysteresis, and sample and hold mechanism), a wide range of dynamic simulations has been performed to optimize its parameters for achieving the best balance between power gain and yaw duty. Dynamic simulations for a realistic case study with a full-scale wind farm indicated 
[revised manuscript text omitted]
 Related to that is the work of Annoni et al. (2019) focused on constructing consensus wind direction estimates.

Combining these techniques with wake steering was considered recently in the work of Sinner et al. (2021) using a modified FLORIS model. Finally, in Kanev (2020) some initial results with optimizing the parameters of a dynamic AWC have been presented. Although the findings there clearly support the necessity of properly optimizing the dynamics of the AWC algo-

100 rithm, the conclusions there remain of limited value due to the simplified modelling approach employed. More specifically, the stationary ELORIS model was used there, extended with a simple time delay model representing wake dynamics. As such

the stationary FLORIS model was used there, extended with a simple time delay model representing wake dynamics. As such, this model is quite unrealistic as it completely neglects spacial inflow variations. Also missing in the modelling approach there is the impact of the increased turbulence in front of waked turbines on their wind measurements, and the resulting increased turbine vawing at downstream turbines.

**105 **Contribution of this work**

The present work extends on above mentioned research in focusing on dynamic robust AWC. The first part of the study concerns *robust* design and analysis using stationary simulations with FarmFlow, a 3D parabolized Reynolds-averaged Navier Stokes code with prescribed pressure gradients and  $k - \epsilon$  turbulence model (Bot and Kanev, 2020). The starting point is the selection of varying or uncertain quantities that can affect the performance of the AWC. To this end, an uncertainty quan-

- 110 tification analysis has been performed for a range of variables (wind speed, wind direction, yaw error, turbulence intensity, wind shear, air density, power curve, thrust curve, power loss coefficient due to yawed error). This analysis indicated that the highest uncertainty contributors are the wind direction, yaw error, turbulence intensity and the wind velocity. Subsequently, Based on the results from this analysis, robust yaw misalignment set-points have been optimized with respect to uncertainties in these parameters the most significant uncertainty sources, modelled as independent stochastic processes with selected PDFs.
- 115 A stationary analysis based on stochastic averaging indicated that the has been carried out to evaluate the performance of the robust AWC design slightly outperforms as compared to the nominal one in terms of power gain.

With the robust AWC in place, the second part of this study continues with the *dynamic* design and analysis. To this end, the originally stationary FarmFlow wake model has been extended to enable dynamic simulations, including wake dynamics and a dynamic yaw control model. The dynamic simulation model is fed by a realistic wind field including temporal and

- 120 spacial inflow variations, that include both micro-scale (fast turbulence variations ranging up to several hundreds of meters) and meso-scale (slow variations extending to ten kilometres and more), with corresponding coherence functions and cross power spectra that relate the stochastic properties between different points in space, and including terms to model the flow advection. The yaw dynamics are modelled at a faster sample rate than the wake model, and an additional higher frequency stochastic signal is added to the yaw error to model the increased noise in the yaw error measurements that enter into the yaw
- 125 position controller. The size of this added noise is made dependent on the turbulence intensity in the flow in front of the turbine. This gives rise to an increased yaw activity of downstream wind turbines, as seen in real-life measurements. Next, an AWC dynamic adaptation algorithm is considered, consisting of a low-pass (LP) filter, a hysteresis, and sample and hold mechanism, similar to (Kanev, 2020). Numerous dynamic simulations have been performed with different dynamic adaptation parameters, both with nominal and robust AWC yaw set-points. Based on the results from these simulations, the optimal parametrization
- 130 of the dynamic AWC are determined. The resulting dynamic robust AWC is shown to deliver a large reduction in the yaw duty in combination with increase in the power gain as compared to the nominal AWC solution.

In summary, the main contributions in this work are as follows:

- Development of dynamic wake model, based on the originally static FarmFlow tool, suitable for design and evaluation of AWC solutions.
- Exhaustive uncertainty quantification analysis, pinpointing the most important uncertainty contributors that need to be considered in a robust AWC design setting.
  - Optimization of the parameters of a dynamic AWC algorithm using a wide range of dynamic simulations, with the purpose of maximizing the power gain and minimizing the yaw duty.
  - Design and evaluation of a dynamic robust AWC algorithm for a realistic case study with a full scale wind farm.

**140 Structure of the paper**

The remaining part of the manuscript is organized as follows. The next section summarizes the modelling used in the study. Sect. 3 outlines the the uncertainty quantification analysis, the selection of the dominant uncertainties, the design of robust AWC with respect to these and, finally, gives the results from a stationary robust analysis. Next, the optimization of the AWC dynamic adaptation algorithm is discussed in Sect. 4. Sect. 5 goes on with demonstrating the benefits from the developed dynamic robust

145 AWC methodology on a case study with a model of an existing offshore wind farm. The manuscript is concluded with some final remarks in Sect. 6.

**2 Wind farm Simulation model description**

The wind farm simulation model consists of a stochastic wind field generator, dynamic wake model (dynamic FarmFlow), model of the wind turbines' yaw systemsmodel, and dynamic robust AWC. A block scheme of the simulation model is given in

- 150 Fig. 1. The, in which the four mentioned main components have been clearly indicated. The yaw models and the dynamic robust AWC are implemented in a dynamic link library (DLL), which is called by the dynamic FarmFlow code at each simulation step. The different line colors are meant to indicate different sample times. The base sampling rate (black lines) is set by the wind field time series (typically 0.1Hz 0.1 Hz or slower). The yaw model operates at faster sampling rates (green lines) to enable realistic modelling of the yaw motion (e.g. 1Hz 1Hz), which is important to assess the impact of AWC on the yaw duty. The
- 155 actual simulation model has a much more extensive interface between the wake model and the AWC algorithm enabling a wide range of possible future applications, but is visualized here in a simplified way, sufficient for the present discussion. Finally, part of the AWC algorithm may operate at slower sampling rates (red lines). The main components are explained separately in more detail in the remainder of this section.

**2.1 Wind field**

160 A stochastic wind field generator is created that produces two-dimensional wind fields with spatiotemporal variations, enabling dynamic wind farm simulations for analysis and design of wind farm control strategies. Control-oriented wind farm simulations

Figure 1. Block scheme of the simulation model

require much slower time scales (tens of seconds) than the aerodynamic simulations needed for evaluating wind turbine controls (which are typically in the tens to hundreds of milliseconds range). Moreover, wind farm simulations require wind fields that extent beyond the traditional duration in wind turbine simulations, usually limited to ten minutes wind fields with a spacial

- 165 range of up to several hundred meters (micro scale). The size of some current wind farms extend to ten kilometres or even more (meso scale). The approach to wind field generation followed in this work is based on modelling of both micro and meso scale effects. The chosen spacial resolution of the wind fields is around two rotor diameters (2D), while the which is in accordance with the wake meandering modelling in Larsen et al. (2008). The sample time is in the order of 10-30 s (roughly equal the time it takes air flow to cover a distance of 2D). Notice that this approach is quite similar to those followed by Bossanyi (2018) and
- 170 Simley et al. (2020), where the authors also split the wind field spectrum into low-frequency (for the wake dynamics) and high-frequency (for the turbine yaw dynamics). In Bossanyi (2018), the same spacial resolution is used for the low-frequency wind field as well.

[revised manuscript text omitted]

---

## Author Comment (AC2)

**Response to Reviewer RC1**

Dear Bart, thank you so much for the kind words, the thorough review of our manuscript and the numerous suggestions for improvements. Below, we have listed your comments and have provided our response after each one. The changes made to the manuscript are indicated by the boxed texts.

**Major comments:**

1. The most significant improvement for this manuscript would be an increased clarify and structure, in my eyes. Often, sections are very long and one loses track of the purpose of a section. I would very much like to see the text restructured into more subsections and paragraphs. For example:

    a. Introduction: separate subsections for modern challenges in large-scale implementation.
    **Response**: Thanks for the suggestion. We have added the following headings in the Introduction section: AWC implementation challenges, State of the art, Contribution of this work, Structure of the paper

    b. Introduction: too, much information on the dynamic FarmFlow part. I think you can remove lines 103-110: from "the dynamic simulation..." until "...in real-life measurements." without losing any valuable information in the introduction.
    **Response**: We have removed these lines from the manuscript.

    c. Section 2.3, page 8: why is the derivation of wind direction variability part of the "yaw model"? This, to me, should be part of the inflow/wind field model.
    **Response**: A good point! The organization of Section 2 was not clear enough, and we have now made an attempt to clarify the structure. To this end, Figure 1 has been updated to clearly indicate the four main components of the simulation model: wind field generator, dynamic wake model (dynamic FarmFlow), wind turbines' yaw model, and dynamic robust AWC. The yaw models and the dynamic robust AWC are implemented in a DLL, which is called by the dynamic FarmFlow code at each simulation step. Therefore, the yaw model is not part of the FarmFlow model (as one might probably expect), but is implemented separately in the DLL. Same holds for the added noise to the wind direction signals, coming from FarmFlow, which is included to model the effect of the increased turbulence in the wake on the measured wind direction. To clarify the structure of Section 2, next to the clarifications made in Figure 1, we have modified the name of Section 2 to "Simulation model description" and have updated the text in the beginning of Section 2 as follows:

    > **"**The wind farm simulation model consists of a stochastic wind field generator, dynamic wake model (dynamic FarmFlow), wind turbines' yaw model, and dynamic robust AWC. A block scheme of the simulation model is given in Fig. 1, in which the four mentioned main components have been clearly indicated. The yaw models and the dynamic robust AWC are implemented in a dynamic link library (DLL), which is called by the dynamic FarmFlow code at each simulation step...
    > ...The main components are explained separately in more detail in the remainder of this section."

d. Table 1: this is a table related to validation of the model choices. This seems somewhat out of place to me, since you are still explaining the fundamentals of the model.
**Response**: The table does seem somewhat misplaced here, indeed, but we could not find any better place for it. It is included to support the idea of adding turbulence-dependent noise term in the wind direction measurements entering the yaw model, resulting in increased yawing of downstream turbines. We could remove the table and the related text, but we do believe it adds value to the discussion. Please, feel free to make a specific advice as to where we could better place the table.

e. Section 2.4: why is this part of Section 2: "wind farm model"? Typically, the wake/wind farm controller is not considered to be part of the wind farm model, especially in this entire context. Perhaps instead this should become part of Section 3 and Section 3 should become "AWC design"
**Response**: We believe this comment is much related to comment (c) above, i.e. the lack of clarity of the structure of Section 2. As explained in our response to comment (c) above, the structure has been clarified in the revised manuscript. Section 2 describes the complete simulation model which consists of four main components, as depicted in Fig.1. These are wind field generator, dynamic wake model (dynamic FarmFlow), wind turbines' yaw model, and dynamic robust AWC. Each component of the simulation model is then described in a separate subsection. Section 3 "Robust AWC optimization" is on the optimization of the parameters of the robust AWC (the LuT in Fig. 1), which Section 4 – on the optimization of the dynamic part of the AWC algorithm (LP filter, hysteresis and sampling). To clarify this further in the manuscript, we have included the following text in the beginning of Section 2.4:

"This section describes the structure of the dynamic robust AWC algorithm, represented by the shaded area at the bottom of Fig. 1. The optimization of the parameters of the underlying blocks of this algorithm is topic of Sect. 3 (Robust AWC optimization) and 4 (Dynamic adaptation algorithm optimization)."

f. Section 3.1: find a way to clearly separate each factor of uncertainty/parameter. Perhaps a bullet point list or subsections/paragraphs.
**Response**: we have improved the structure to Section 3.1 in the revised manuscript by using a list.

g. Line 437: can start a new subsection *(see next comment)
**Response**: see response to next comment.

h. Latter half of Section 4 vs. Section 5. One shows a basic case study for 3 turbines, and the other shows a more realistic case study with OWEZ. To me, it would make sense to put them both in Section 5 and separate them into two subsections: one for verification/simple study case for understanding, and then one for a more realistic evaluation.
**Response**: Good idea! We moved the simplified example with 5 turbines from Section 4 to the beginning of Section 5, and making Section 5.1 out of it. The realistic case study with OWEZ became Section 5.2. We added the following text to the beginning of Section 5 to classify this:

> "This section presents the results from two case studies. The first one represents a example of robust AWC design performed for a simple farm consisting of a few turbines in a row. The second one represents a realistic case study with dynamic robust AWC applied to an offshore wind farm."

i. I read that FarmFlow has been extended to include wake and yaw control dynamics. It also now accepts dynamic wind fields to drive the simulation, including temporal and spatial variations. These are great developments. I would very much appreciate any kind of validation of these new functions. However, with the paper already being as long as it is, perhaps it would be better to present the dynamic FarmFlow plus validation in a separate publication. This would also increase clarity in the current manuscript.

**Response**: Very valid point. The dynamic part of the FarmFlow model has, unfortunately, not been properly validated yet, and we are currently looking for funding to support this very important work. Some sanity checks have, of course, been done to ensure the output makes sense, but a more rigorous validation is still needed. Once this is done, we will of course consider publication.

2. Please motivate certain statements with the right literature and avoid speculation

   a. In the introduction, I read that the potential AEP gain with AWC is several percents. This seems very high and currently not too realistic based on the recent expert elicitation and field experiments that exist in the literature. Actually, the papers cited with this statement are simulation studies and are better replaced with the Wingerden et al. expert elicitation and field experiments from Howland, Fleming, Simley, Duc and Doekemeijer. This relates to the minor comment on citing literature.

   **Response**: Several percent AEP increase is not stated anywhere in the manuscript. Instead, both in the Abstract and in the Introduction a *possible, or potential, gain of up to a few percent* is mentioned. In my opinion, this is not exaggerating the current knowledge. Indeed, this statement is only backed up by numerical studies, but the field experiments are currently only very limited and provide no basis for estimation of the AEP gains achievable in the future. Furthermore, the paper on expert elicitations does not cover the question of what is the expected AEP increase, but rather the following one: "How much of an increase in energy production is needed to justify implementation?". In our opinion, due to lack of other evidence, we believe it is better to stick to our initial citations.

   b. Line 33, where the author assumes that the accumulated loads over the whole lifetime of a wind turbine decreases with wake steering, rather than increases, because Siemens-Gamesa is selling a wake steering solution. This reasoning seems flawed to me. We do not fully know under what conditions Siemens-Gamesa is doing wake steering, if they require additional equipment, whether and which loads increase and decrease, when they do, and whether this relates to fatigue or ultimate loads. There is too little information to make any conclusions based on the fact that Siemens-Gamesa is selling wake steering, besides perhaps that it has caught the interest of this OEM.

   **Response**: It is not true that we assume that the lifetime fatigue loads decrease with wake steering because SGRE is selling such a solution. Instead, the manuscript

clearly states: "*A more detailed study involving a utility scale wind farm was presented in Kanev et al. (2020), where the impact of wake steering AWC on the structural loads of the turbines during their complete lifetime has been investigated using the so-called loads lookup table (LUT) approach (Reyes et al. 2020). The results demonstrate that, even though by itself yaw misalignment does increase the structural loads of some turbines in specific wind conditions, the wake-induced loading is decreased even more, so that the accumulated loads over the whole lifetime of each wind turbine generally remain lower than without AWC.*".
Nevertheless, we have removed the following sentence in the revised manuscript to avoid any misinterpretations:

> "This conclusion is implicitly confirmed by the fact that the industry starts to develop this technology into commercial products (Siemens Gamesa Renewable Energy, 2019)."

c. Line 134-135, it is stated that wind farm simulations require time scales of tens of seconds. How about wake meandering or finer flow effects? How about large-eddy simulations? Add citations or at least defend this statement. Similarly, motivate choices of spatial resolution and sample time of the inflow.
   **Response**: The wind farm modelling required for this study needs only to include effects necessary for modeling the power production of the wind turbines, the wake meandering, and the yaw dynamics. Higher frequencies (>1Hz) are not correlated across the wind farm and represent local turbulence variations. These are important for the turbine loads, of course, but loads modeling falls outside the scope of the model. Higher frequencies are, however, relevant for the proper modeling of the nacelle yaw dynamics, and are therefore included into the model (but uncorrelated in space). The modelling approach, followed in this work, is actually quite well aligned with earlier work of others: Bossanyi (2018) and Simley (2020). We have now indicated this in the revised manuscript through the following addition at the end of the first paragraph of Section 2.1:

> "Notice that this approach is quite similar to those followed by Bossanyi (2018) and Simley et al. (2020), where the authors also split the wind field spectrum into low-frequency (for the wake dynamics) and high-frequency (for the turbine yaw dynamics). In Bossanyi (2018), the same spacial resolution is used for the low-frequency wind field as well."

Regarding the sampling time, the text states already that it is

> ".... roughly equal the time it takes air flow to cover a distance of 2D."

Given the fact that 2D is the special resolution, we believe this makes sense.

d. Line 302: motivate natural frequency of meandering
   **Response**: We could not find the text (around line 302) to which this comment relates. However, for the good order, we added a relevant citation in the first paragraph of Section 2.1 in the revised manuscript, regarding the wake meandering modeling on which our choice of 2D for the special resolution is based:

> "The chosen spacial resolution of the wind fields is around two rotor diameters (2D), which is in accordance with the wake meandering modelling in Larsen et al. (2008)."

e. Would be nice to clearly define the novel contributions in this article vs. what was done in previous work. FarmFlow already existed, but has been made dynamic: that is new, no? Uncertainty quantification is novel, at least for that exhaustive of a parameter set. Robust AWC and hysteresis already existed in literature, right?
**Response**: Good point. As already mentioned in our response to 1 a), we added a heading "Contribution of this work" in the introduction. We have now also added a summary of the main contributions of the paper for more clarity:

> **"**In summary, the main contributions in this work are as follows:
> - Development of dynamic wake model, based on the originally static FarmFlow tool, suitable for design and evaluation of AWC solutions.
> - Exhaustive uncertainty quantification analysis, pinpointing the most important uncertainty contributors that need to be considered in a robust AWC design setting.
> - Optimization of the parameters of a dynamic AWC algorithm using a wide range of dynamic simulations, with the purpose of maximizing the power gain and minimizing the yaw duty.
> - Design and evaluation of a dynamic robust AWC algorithm for a realistic case study with a full scale wind farm."

**Minor comments**

1. The abstract contains the general outline of the paper but misses the actual contributions and results. It currently does not suffice as a standalone summary of the paper. Please include the core findings, qualitatively but also quantitatively. For example, depict the parameters that were found to be the most important from the sensivity analysis, depict the potential AEP gain in percent, and so on.
**Response**: We have extended the abstract as suggested, by including the following text:

> **"**To this end, an uncertainty quantification analysis has first been performed for a range of variables (wind speed, wind direction, yaw error, turbulence intensity, wind shear, air density, power curve, thrust curve, power loss coefficient due to yawed error), which indicated the wind direction, yaw error, turbulence intensity and the wind velocity as the highest uncertainty contributors. Robust AWC has next been synthesized by including stochastic uncertainties in these parameters. A stationary analysis through stochastic averaging indicated that the robust AWC design only slightly outperforms the nominal one in terms of power gain. For the dynamic design and analysis, the originally stationary FarmFlow wake model has been extended to enable dynamic simulations, including wake dynamics and a dynamic yaw control model. By selecting a certain dynamic adaptation algorithm structure (a low-pass filter, hysteresis, and sample and hold mechanism), a wide range of dynamic simulations has been performed to optimize its parameters for achieving the best balance between power gain and yaw duty. Dynamic simulations for a realistic case study with a full-scale wind farm indicated that the developed dynamic robust AWC results in a large reduction of the yaw duty (30-50% lower) while at the same time improving the overall power gain (2.05% vs. 0.56%), as compared to the conventional nominal AWC."

2. Generally, and especially when citing literature, you should clarify the test environment used in that publication. The differences between a FLORIS-based simulation study, a SOWFA-based simulation study, a field experiment or a wind tunnel experiment are very significant.

**Response**: Of course, and we believe we have done that in many places in the original manuscript, such as on Line 55 "…in recent field studies with wake redirection (Fleming et.al., 2020, 2019)", Lines 67-69 "Above-mentioned studies on robust AWC were all performed using a simplified control-oriented wake model, namely the FLOw Redirection and Induction in Steady State (FLORIS) model - an understandable choice given the computational requirements for robust optimization.", Lines 70-71 "…utilizing a different steady-state wake model, called the lifting line model.", Line 83 "In Simley et.al (2020), for instance, dynamic simulations have been performed using the stationary FLORIS wake model", etc. It is unclear to us what the point of this comment is.

3. For literature review: similar work is from M. Sinner et al., 2021, but this only appeared in April 2021. I can understand that the authors had already finished this publication mostly by then. You could consider including it in a revision.
   "Power increases using wind direction spatial filtering for wind farm control: Evaluation using FLORIS, modified for dynamic settings", Sinner et al., 2021, JRSE
   **Response**: Thanks for pointing us out to this relevant recent publication, we have of course included a citation in the revised manuscript through the following text in the introduction:

   > "Combining these techniques with wake steering was considered recently in the work of Sinner et al. (2021) using a modified FLORIS model."

4. Line 96: "This analysis ... the wind velocity." This is a conclusion and should not be part of the introduction. Rather, the introduction should be limited to what topics will be addressed in the article. The same goes for the sentence starting at
   line 99: "A stationary analysis ... of power gain." Nice, but should go to conclusion.
   **Response**: Agreed. The sentence on Line 96 of the original manuscript has been removed, and the next sentence modified to:

   > "Based on the results from this analysis, robust yaw misalignment set-points have been optimized with respect to the most significant uncertainty sources, modelled as independent stochastic processes with selected PDFs."

   The sentence commencing at Line 99 has been modified to:

   > "A stationary analysis based on stochastic averaging has been carried out to evaluate the performance of the robust AWC design as compared to the nominal one in terms of power gain."

5. Line 182: "wake generated by a wind turbine is propagated downstream based on the local wind direction variations in its way", I do not understand this.
   **Response**: The sentence has been rephrased as follows:

   > "To this end, the wake generated by a wind turbine is propagated downstream in such a way that it follows on its way the local wind direction variations in the wind field. This way, both time delay and meandering effects are modelled."

6. Line 183: "because the traveling time ... current time window." I do not understand this.
   **Response**: This sentence has been rephrased as follows:

   > "Since the travel time of a wake between two turbines takes longer than the simulation sample time, …"

7. Line 202: "written in an output file", this seems inefficient. Can this not directly be

exchanged through memory or over a network protocol?

**Response**: This certainly can, and might be implemented in future updates of the software tool.

8. Line 260: On what signal does the LP filter work?

   **Response**: The following clarification was added to the LP filter description:

   > **"**As visualized in Fig. 1, the LP filter acts on the wind speed and wind direction signals.**"**

9. Line 314: You mention that the PDF for turbulence intensity is based on historical data. Does your definition of TI (i.e., being the standard deviation in streamwise direction, match up with the definition in the data? I can imagine that the historical data considers the TI to include both streamwise and cross-stream turbulence.

   **Response**: Our definition of TI in the wake model and in the wind field generator is, in fact, pretty standard, and as such are strictly speaking not exactly matching the definition TI measurements based on 10 minute statistical met mast data (mean value and standard deviation of the measured wind velocity). However, notice that these data is used to construct a rough, though realistic, statistical model of the turbulence intensity *variations*, which serves primarily as an example to based the robust analysis and design on. We do not believe that tinkering around the edges here will improve the value of the paper.

10. Line 349: What optimization algorithm is used? How confident are you that the solution has converged? What are the bounds, e.g., have you limited the minimum and maximum yaw angles?

    **Response**: The following text has been added in Section 3.2 to explain the optimization algorithm used in this study:

    > **"**To solve the underlying optimization problems, a tailor made algorithm has been used that requires a minimum number of function evaluations (farm simulations) to converge. The algorithm is similar to the conventional bisection method, but generalized to multivariate objective functions. By confining the optimization variable to lie within an initial n-dimensional box, the gradient of the objective function is evaluated at the centre point at each iteration and the box is reduced in size by keeping only that part that is oriented opposite to the gradient. While this algorithm has no theoretical guarantees to converge to an optimum solution for general nonlinear functions, has been successfully used for many years by the authors and works pretty well for the application at hand, its low calculation effort being its main advantage over alternative algorithms. This allows it to be used in combination with relatively complex wake models such as FarmFlow. To reduce computation time even more, the number of optimization variables is limited to the yaw set-points of the two most upstream turbines in each row of turbines oriented downstream. The yaw set-points for the remaining turbines in the row are linearly decreased between the second turbine and the last one, which has zero yaw misalignment set-point. No limitation has been applied to the yaw set-points in this section.**"**

    To clarify that this algorithm is used in all optimizations throughout the paper, the following like is added at the end of Section 3.3:

    > "All optimization problems are solved by using the algorithm, described in Sect. 3.2. The yaw misalignment angles have been limited to ±30 degrees."

11. Table 3: the wind speed range seems so high, while in reality you could feed in the wind speed measurements into the LUT, perhaps with an uncertainty bound but definitely smaller than an uncertainty of 8 m/s. How do you defend this decision? Also, how do these findings line up with your earlier work stating that wind speed can be ignored in yaw optimizations?
**Response**: In fact, the idea is to not use the wind speeds as an argument for the LUT (other than for switching AWC on and off), but rather to have the LUT robust with respect to the whole range of wind speed variations. This is mentioned in the first paragraph of Section 3, as well as in the second bullet point in Section 3.1, where the modeling of the wind speed uncertainty is described. Nevertheless, to better clarify this, the following line is added to the mentioned bullet point:

> **"**Instead, the LUT will be designed to be robust to wind speed variations.**"**

12. Line 381: I would have expected the yaw-induced power loss coefficient to have a larger effect on the optimal yaw angles, since it directly impacts the energy lost by yawing an upstream wind turbine. Can you reason why this is not the case in this study?
**Response**: You are absolutely right. Large variations in the yaw-induced power loss coefficient result in significant variations in the optimal yaw set-points. Take, for instance the result for Case 2 in Figure 5, depicted by the middle (green) bar plot. The optimal yaw misalignment of the first turbine is around 22° for yaw-induced power loss coefficient of 2.3, and 35° for a coefficient of 1.3. Computing the corresponding power losses, $\cos(\beta)^{\alpha}$, one gets 0.84 for $\alpha$=2.3 and $\beta$=22°, and 0.77 for a $\alpha$=1.3 and $\beta$=35°. The larger power loss at the first turbine in the later case is then compensated by the corresponding higher yaw misalignment, leading to increased power production downstream. So the results make perfect sense. Including the assumed distribution in Figure 4, however, narrows down the most probable range of variations of the yaw-induced power loss coefficient to below 1.7, leading to a quite small size of the green box in Figure 5. To summarize, the yaw-induced power loss coefficient does affect the optimal yaw set-points, but for the assumed uncertainty model (PDF) this impact remains limited and the parameter does not need to be considered in the robust optimization.

13. Figure 5: yaw angles of -40 deg and + 45 deg seem excessive. Can you explain your choice for allowing yaw angles to go all the way to these values? Since we would never optimize the yaw angles until those limits in practice, this may skew the sensitivity analysis somewhat, no? Perhaps certain parameters are important at high misalignment angles, but really are not that important in the range we expect to yaw the turbines to.
**Response**: This is, in fact, quite a good point, and not touching upon it is indeed an omission. The optimizations in Section 3.2 have been consciously performed without applying limitations on the yaw misalignment, as the exact limitations that will apply in future applications are not exactly known and the results from the uncertainty quantification analysis should be generic. However, as limitations will apply in practice, we agree it is important to discuss their impact. We have added the following paragraph to Section 3.2 to discuss this important aspect:

> **"**Notice that, due to the fact that no limitations have been imposed, the yaw misalignment set-points are getting quite high in some cases, raising to values of 40 degrees and even higher. Such a high yaw misalignments are currently considered unrealistic in a real-life application. Usually, they are limited to around 30 degrees in many research studies, or to even lower values in the first field tests with wake redirection (Flemming et al., 2017, 2019;

Doekemeijer et al., 2021). It becomes clear from Fig. 5 that by imposing a limitation of ±30 degrees one would significantly limit the variation in the yaw misalignment set-points. Nevertheless, the main conclusions drawn above in terms of the most significant uncertainty contributors will still hold, with probably only the wind shear disappearing from this list.**"**

14. Figures 5 & 6: please add legends:
    **Response**: Legends added to Figures 5 and 6!

15. Line 414: should it read 'arg max' instead of 'arg min'?
    **Response**: Of course it should. We have fixed this, as well as 4 other occurrences!

16. Line 434: "...only decent directions...", what are "decent directions"?
    **Response**: This sentence is not precent in the revised manuscript. Instead, as explained in the Response to question 10 above, the description of the optimization algorithm has been explained in detail in Section 3.2.

17. Figure 8: neither line is particularly smooth. Does this suggest that the optimization has not converged?
    **Response**: This has to do with the nonlinearily of the objective function, the chosen optimization algorithm, and the termination criterion, indeed. It is, of course, possible to improve the optimizer to get smoother curves, but that would require increasing the calculation time significantly. We believe, however, that this is not really necessary since resulting power gain is not very sensitive on such relatively small variations of the yaw misalignment angles. This fact can be appreciated from Figure 12, noticing that the power gain barely changes between nominal and robust AWC. Smoothing of the yaw set-points can be easily done after the optimization, which does save a lot of calculation effort, especially when designing AWC for the complete spectrum of wind conditions in a full scale wind farm. Robust design does, however, deliver smoother yaw angles in general.

18. Figure 9: "robust AWC" and "nominal AWC (with uncertainty)" are not the same thing, yet it is hard to distinguish them in their definitions. Can you clarify?
    **Response**: We have added the following text at the end of Section 5.1 to clarify the different curves in the figure (now Figure 12 in the revised manuscript due to the change explained in the Response to Major comment 1h):

    **"**For the sake of clarity, the difference between the red curve "nominal AWC (no uncert.)" and the black curve "nominal AWC (with uncert.)" in Fig. 12 is that the former one depicts the power gain evaluated just for the nominal values of the uncertainty parameters ($p^\wedge nom$), while the later one represents the gain evaluated by including the whole uncertainty set $U$ through the joint PDF $D\_d$ ($p$). In both cases, the yaw misalignment set-points are the same, namely $\gamma\_nom$, optimized for the nominal values of the parameters $p^\wedge nom$, i.e. neglecting the uncertainty, as defined in Eq. (4). The blue curve "robust AWC" corresponds to the case when both the optimization of the yaw set-points and the evaluation of the power gain are performed by accounting for the uncertainty through $D\_d$ ($p$)."

19. Figure 11: 11% Energy gain is very substantial and not particularly realistic for AEP. Maybe repeat that this is for particular 3-turbine case. Also, these figures are hard to see. I would suggest turning them into top-view (2D) contour plots instead. Same goes for Figure 12.
**Response**: The 11% gain is not for AEP but for the performed simulation conditions only, mean wind velocity of 8 m/s and wind direction varying around 270 degrees. 11% power gain is not at all unusual in such a scenario, as you would agree. We added a clarification about the simulation conditions in Section 4 as follows

> **"**The average wind velocity is 8 ms$^{-1}$, and the wind direction varies around 270° (see light grey line in Fig. 10).**"**

20. Line 555: Just to clarify, so dynamic FarmFlow runs 1:1 (6 hours of simulation means 6 hours of computing in real time on a single core)? If so, it may be worth evaluating the potential for a full year of operation (~9,000 CPU hours).
**Response**: Correct. A full year can easily be calculated on a computer cluster.

21. Line 567: "Notice that the overall gains are lower than one might expect", what would be a reasonable number to expect? 0.5-2% energy gain is still significant if you ask me.
**Response**: Again, the power gains are not on annual basis (AEP), but for the simulated wind conditions only. You would agree that for the "more beneficial" wind directions the power gain is usually higher than that.

**Technical comments**

1. Variables should be italic, units should not.
**Response**: We have carefully gone through the text, including the tables, and changed all noticed occurrences of italic units to regular. The variables were properly typeset in the original manuscript already.

2. Line 4: "by up to a few percentage points." Why percentage points and not percent?
**Response**: "percentage points" changed to "percent".

3. Line 19: "possible power gains of up to a few percent on annual basis". Can you motivate this further, maybe add citations? To me, it seems that it is more towards a single percent, especially when looking at the most recent field experiments.
**Response**: This question is quite similar to that posed in Major comment 2a. Please, refer to our response to that comment.

4. Line 20: The second challenge is presented as being mainly due to wake models being of static nature.
**Response**: As stated in the text: "The second challenge is related to the uncertainty in the predictions for the expected annual energy production (AEP) increase, caused by the simplistic static approach that is currently used to optimize AWC on the one side, and the underlying uncertainties in the modelling used for that purpose on the other side."

5. Figure 1: it says "yaw systems". Should it be "yaw system" since its for a single turbine?
**Response**: Yes, we changed it to "yaw system".

6. Figure 1: The Robust AWC LUT seems only a function of wind direction and wind speed. Does this mean local WS/WD?
   **Response**: The local wind conditions are used by the AWC algorithm. To clarify this in the text, the following line is added in the first paragraph of Section 2.4:

   > **"**In the implementation, used in this work, the AWC algorithm receives the local wind speed and wind direction as measured at turbine level.**"**

7. Line 149: "frequencies above 10e-3 Hz", should this be "above 10e-3 Hz"?
   **Response**: I don't understand this comment. I believe the sentence is clear: "This Kaimal spectrum is used for frequencies above $10^{-3}$ Hz, i.e. time scales of 30 minutes and slower."

8. Line 279: "In this work ... robust design setting." I understand what you mean, but perhaps reformulate it in a clearer way. For example, differentiate between variables included as uncertainties in the optimization process and variables that are used for the real-time interpolation of setpoints.

   > **Response**: We have rewritten this text as follows to improve the clarity:
   > **"**In this work, the yaw misalignment set-points in the LUT will essentially be a function of the wind direction only. The wind speed signal entering the block "Robust AWC (LUT)" in Fig. 1 is meant to indicate that AWC is only active in a certain range of below rated wind conditions (4-12 ms$^{-1}$ used here).**"**

**Response to Reviewer RC2**

Dear reviewer, thank you so much for the kind words and the useful comments and suggestions for improvements. Below, we have listed your comments and have provided our response after each one. The changes made to the manuscript are indicated by the boxed texts.

**General Comments:**

- Section 2.1: Do I understand that wind directions variations occur in the range of 30min - 24hr, and that faster frequencies are uncorrelated spatially? If we expect a wind turbine to yaw something like several times every 10 minutes does this match?
  **Response**: No, in fact both the micro-scale model ($f > 10^{-3}$ Hz) and the meso-scale model ($f < 10^{-3}$ Hz) include special correlation, as modelled in equation (1). Kindly note the sentence just before the equation stating "For the complex cross power spectrum between two points in space, $r$ and $s$, the following expression is used for both the micro and the meso scale spectra …".

- Could you provide a definition of stochastic programming in general and how it is used in this work?
  **Response**: Definition of the stochastic programming problem, as well as explanation of how it is numerically solved in our work, are explained in detail in Section 3.3. Since the objective function cannot be evaluated in its continuous form, the joint PDF is first discretized and the resulting optimization problem is solved through the algorithm described in Section 3.2. This approach, based o discretization, has already been used by others in this application, and we have included citation to that work in the revised manuscript:

> "To solve this problem numerically, the continuous PDFs are discretized as in Rott et al. (2018); Simley et al. (2020)."

**Specific comments:**

- Page 2: "This conclusion is implicitly confirmed by the fact that the industry starts to develop this technology into commercial products (Siemens Gamesa Renewable Energy, 2019)." Could also be that the loads are higher but not importantly so?
  **Response**: This is certainly true. The point we were trying to make is that the loads don't seem to be an obstacle for the implementation of AWC, but we have removed the following sentence in the revised manuscript to avoid any misinterpretations:

> "This conclusion is implicitly confirmed by the fact that the industry starts to develop this technology into commercial products (Siemens Gamesa Renewable Energy, 2019)."

- Page 3: "In a different work, the same author demonstrates that a centralized yaw control strategy, in which information from surrounding wind turbines is used in the yaw control algorithm, can lead to a drastic reduction in the yaw duty and increase the power capture at the same time (Bossanyi, 2019)." This could also be related to the concept of consensus control: Annoni, J., Bay, C., Johnson, K., Dall'Anese, E., Quon, E., Kemper, T., and Fleming, P.: Wind direction estimation using SCADA data with consensus-based optimization, Wind Energ. Sci., 4, 355–368, https://doi.org/10.5194/wes-4-355-2019, 2019.
  **Response**: This is definitely a very relevant work, we have included a reference to it in the Introduction through the following text:

> "Related to that is the work of Annoni et al. (2019) focused on constructing consensus wind direction estimates."

- This sentence: "This Kaimal spectrum is used for frequencies above 10−3 150 Hz, i.e. time scales of 30 minutes and slower". If the range is above a frequency, do you mean lower and not slower?
  **Response**: Yes, slower is now changed to lower.

- Page 6: "The parameter c(αrs) is the decay factor". Decay of what?
  **Response**: We have added the following clarification regarding this parameter:
  **"**The parameter $c(\alpha_{rs})$ is the decay factor, a parameter characterizing the decay rate of the coherence function.**"**

- Page 10: Recommend to explain figure 2 in more detail in the caption
  **Response**: We modified the caption of the figure as follows to make it more explanatory:

  **"**Illustration of the impact of wake on wind measurements and yaw motion of two commercial wind turbines, when the second turbine (T2) is in the wake of the first one (T1). The thin lines represent LP filtered wind direction measurements at the two turbines, while the think ones -- the nacelle position measurements.**"**

- Page 13: Don't need to revise the paper, but wanted to note I think some recent papers might point to a distribution for yaw loss exponent centered somewhat higher, or even dependent on wind speed: Simley, E., Fleming, P., Girard, N., Alloin, L., Godefroy, E., and Duc, T.: Results from a Wake Steering Experiment at a Commercial Wind Plant: Investigating the Wind Speed Dependence of Wake Steering Performance, Wind Energ. Sci. Discuss. [preprint], https://doi.org/10.5194/wes-2021-61, in review, 2021.
  **Response**: Of course, very relevant work. We have included a reference to it in the text as follows:

  **"**Related to that is the work of Annoni et al. (2019) focused on constructing consensus wind direction estimates.**"**

- Page 15. "Variations in the thrust curve and the yaw-induced power loss exponent have generally limited impact on the optimal yaw set-points, which suggests that they could be left out from the robust optimization." This is surprising, at least for the power curve exponent, it would seem that at some loss level it would start to have a strong impact,?
  **Response**: You are absolutely right. Large variations in the yaw-induced power loss coefficient result in significant variations in the optimal yaw set-points. Take, for instance the result for Case 2 in Figure 5, depicted by the middle (green) bar plot. The optimal yaw misalignment of the first turbine is around 22° for yaw-induced power loss coefficient of 2.3, and 35° for a coefficient of 1.3. Computing the corresponding power losses, $\cos(\beta)^{\alpha}$, one gets 0.84 for $\alpha$=2.3 and $\beta$=22°, and 0.77 for a $\alpha$=1.3 and $\beta$=35°. The larger power loss at the first turbine in the later case is then compensated by the corresponding higher yaw misalignment, leading to increased power production downstream. So the results make perfect sense. Including the assumed distribution in Figure 4, however, narrows down the most probable range of variations of the yaw-induced power loss coefficient to below 1.7, leading to a quite small size of the green box in Figure 5. To summarize, the yaw-induced power loss coefficient does affect the optimal yaw set-points, but for the assumed uncertainty model (PDF) this impact remains limited and the parameter does not need to be considered in the robust optimization.

- Page 19: "The yaw set-points for the remaining turbines in the row are linearly decreased between the second turbine and the last one, which has zero yaw misalignment set-point." This is a great idea! Is this novel to this paper or has it the concept been used elsewhere?
  **Response**: Well, it's not really new, this is an approach we use for years to reduce the computational time for the optimization. It is first mentioned in our work:
  Kanev, S.; Savenije, F. & Engels, W. Active wake control: an approach to optimize the lifetime operation of wind farms, Wind Energy, 2018, 21, 488-501

- Page 21: Metrics are really useful, the power gain per unit yaw travel increase is very interesting, is this also a novelty of this paper or something used in other papers or other contexts
  **Response**: Well, we haven't seen these metrics in other publications, but they are probably also not too difficult to invent when one tries to capture the wish to optimize the balance between yaw duty and power gain into a cost function.

- Page 23: Results for hysteresis are very promising. If 4 deg is both the highest value tested and the best overall, does it suggest 5 deg or more should be considered?
  **Response**: Yes, in fact it does. This should be considered in future studies. We have included the following comment in the revised manuscript:
  > **"**This result also suggests that even higher values for the hysteresis size be  considered in future studies.**"**

- Page 24: "Having significantly less start/stop events in the reference case than with AWC might first seem couter-intuitive, but does happen". Did this sentence mean to say the reverse (49% reduction)?
  **Response**: You are completely right, thanks for noticing. We have corrected the sentence as follows:
  > **"**Having significantly less start/stop events with AWC than in the reference case might first seem counter-intuitive, but does happen.**"**

- Page 28: Recommend to cite paper mentioned above on consensus control
  **Response**: Due to a recommendation of the third reviewer to shorten the Conclusions, we have completely removed the topics for future research from the manuscript, including the part related to consensus control.

**Response to Reviewer EC1**

Dear reviewer, thank you for the useful comments and suggestions for improvements. Below, we have listed your comments and have provided our response after each one. The changes made to the manuscript are indicated by the boxed texts.

**Remarks:**

- Pg 1. "considered as the most potential technology" please add a citation
  **Response**: We don't quite understand this comment, as there were already three citations in the original manuscript: "More specifically, the wake redirection approach to AWC, employing intentional yaw misalignment to steer wakes away from downstream turbines, is currently considered as the most potential technology with respect to power production increase, with possible power gains of up to a few percent on annual basis (Kanev et al., 2018; Gebraad et al., 2017; Fleming et al., 2016)."
  Please, let us know if there is any specific publication that is more suitable to cite here, and we will be happy to include it.

- Pg 3. "is quite unrealistic" also a rather bold statement
  **Response**: We rephrased the sentence "As such, this model is quite unrealistic as it completely neglects spacial inflow variations" as follows

  > **"**However, it completely neglects spatial inflow variations.**"**

- It is "spatial" instead of "special"?
  **Response**: We used the word "spacial", but agree that "spatial" is more commonly used. We have now replaced all occurrences of "spacial" with "spatial".

- Section 2, at the beginning they talk about different sample rates (yaw vs flow) and there they question arises if you can do that. Later in the article they explain that the flow has two time scales (fast and slow).
  **Response**: We have added the following sentence at the beginning of Section 2 in the revised manuscript to clarify this point:

  > **"**This is achieved by superimposing higher frequency noise to the wind directions coming from the wake model.**"**

- Fig. 1. is really nice
  **Response**: Thank you

- Pg. 7. "because" ..."because" (two sentences in a row)
  **Response**: The repetition has been removed by rephrasing the first sentence as follows:

  > **"**Since the travel time of a wake between two turbines takes longer than the simulation sample time, the arriving wakes from upstream turbines need to be time synchronized with the departure of wakes in the current time window.**"**

- Pg. 7. How the steps are explained it is just not clear. It is also not relevant that you are working with DLL's
  **Response**: The text explaining the different steps in the simulation of the wake model has been revised as follows:

> "1. For a given time instant, the simulation starts with the most upstream turbine with respect to the farm-average wind direction, and ends with the most downstream turbine.
> 2. For a given turbine, all arriving wakes (if any), are determined using wake information calculated at previous time instances and stored in memory in Step 5
> 3. From the undisturbed wind field and arriving wakes, a rotor averaged wind speed and wind direction is calculated.
> 4. Given the determined rotor averaged wind conditions, and the yaw position (coming from the yaw system model in Step 8, see also Fig. 1), the yaw misalignment angle is computed. Subsequently, the power production and rotor thrust force are determined, and a wake calculation is started using the (stationary) wake model in FarmFlow for the given turbine.
> 5. The wake is calculated until reaches a downstream turbine (if any). The path of the calculated wake is determined using the time varying wind direction in the undisturbed wind field, resulting in wake meandering. The wake information is then stored in memory, including the time of arrival at the downstream turbine for use in the wake simulation of that turbine at a later time instant in Step 2.
> 6.  The simulation continues with the following most upstream turbine until all turbines have been simulated for the current time instant.
> 7. The input and output data for all turbines are written in an output file, which forms the interface to a dynamic link library (DLL) that implements the yaw system dynamics and the dynamic robust AWC algorithm.
> 8. The DLL is called, which updates the yaw positions of all turbines, and communicates this information back to FarmFlow. The simulation of the next time instant is initiated in Step 1."

- Section 3, this should really be connected to ongoing efforts/frameworks
  **Response**: We have revised the second paragraph in Section 3 as follows to reflect the contribution of the present work to existing one:

> "This section considers the synthesis of robust AWC in which a number of parameters is treated as uncertain, continuing along the lines of previous work by Quick et al. (2020). Here, a more extensive list of uncertain parameters is treated, consisting of quantities related to model parameter uncertainty, measurement uncertainty and input variability, all treated in a unified stochastic framework using PDFs (Sect. 3.1). The impact of these uncertainties on the optimal yaw set-points is quantified through forward uncertainty propagation (Sect. 3.2). To this end, the PDFs are sampled and the underlying simulations are performed using the conventional (static) FarmFlow wake model. This allows to evaluate how much the optimal yaw set-points change as result of variations in the different uncertain parameters within their assumed uncertainty sets, so that it can be decided which uncertain parameters should be included into the robust optimization. In Quick et al. (2020) a similar analysis is performed by focusing on the impact of variations in the sizes of the uncertainty sets instead. Finally, the robust AWC design is performed by solving a stochastic programming problem through discretization of the probability distributions to arrive at a finite number of scenarios (Sect. 3.3), similar to Rott et al. (2018); Simley et al. (2020)."

  In addition, regarding the wind direction variability having the largest impact, the following line is added to the list of observations, made in Section 3.2:

> "This conclusion confirms earlier observations by \citet{Quick:WES20}."

- Pg 22. Figure 11 is not clear, Figure 12 is better (color) but still hard to read

**Response**: We have updated Figure 11 by plotting the surfaces in color. Notice that it has become Figure 9 in the revised manuscript, as result of part of Section 4 having been moved to Section 5 in response to a suggestion of the first reviewer.

- Conclusion: is a conclusion of a research report and should be shortened. It should answer the research question (is there a research question?).
  **Response**: We have significantly shortened the length of the Conclusions by removing the whole last paragraph related to possible topics of future research. In addition, we have stated the research question more clearly in the beginning of the Conclusions as follows:

  > **"**The aim of this paper is to develop a framework for the design of dynamic robust AWC for optimizing the balance between the yaw duty and the power gain under realistic conditions, accounting for wind resource variability and model and measurement uncertainty, as well as to assess its benefits with respect to the conventional nominal AWC design.**"**

  We believe the remainder of the section is clearly focused on this research question, briefly explaining the methodology and focusing on the conclusions drawn from the obtained results.

**Dynamic robust active wake control**

Stoyan Kanev[1] and Edwin Bot[1]

[revised manuscript text omitted]

In summary, the main contributions in this work are as follows:

– Development of dynamic wake model, based on the originally static FarmFlow tool, suitable for design and evaluation of AWC solutions.

135 – Exhaustive uncertainty quantification analysis, pinpointing the most important uncertainty contributors that need to be considered in a robust AWC design setting.

– Optimization of the parameters of a dynamic AWC algorithm using a wide range of dynamic simulations, with the purpose of maximizing the power gain and minimizing the yaw duty.

– Design and evaluation of a dynamic robust AWC algorithm for a realistic case study with a full scale wind farm.

140 **Structure of the paper**

The remaining part of the manuscript is organized as follows. The next section summarizes the modelling used in the study. Sect. 3 outlines the the uncertainty quantification analysis, the selection of the dominant uncertainties, the design of robust AWC with respect to these and, finally, gives the results from a stationary robust analysis. Next, the optimization of the AWC dynamic adaptation algorithm is discussed in Sect. 4. Sect. 5 goes on with demonstrating the benefits from the developed dynamic robust
145 AWC methodology on a case study with a model of an existing offshore wind farm. The manuscript is concluded with some final remarks in Sect. 6.

**2  Simulation model description**

The wind farm simulation model consists of a stochastic wind field generator, dynamic wake model (dynamic FarmFlow),  wind turbines' yaw model, and dynamic robust AWC. A block scheme of the simulation model is given in
150 Fig. 1, in which the four mentioned main components have been clearly indicated. The yaw models and the dynamic robust AWC are implemented in a dynamic link library (DLL), which is called by the dynamic FarmFlow code at each simulation step. The different line colors are meant to indicate different sample times. The base sampling rate (black lines) is set by the wind field time series (typically  0.1 Hz or slower). The yaw model operates at faster sampling rates (green lines) to enable realistic modelling of the yaw motion (e.g. 1 Hz), which is important to assess the impact of AWC on the yaw duty.
155 This is achieved by superimposing higher frequency noise to the wind directions coming from the wake model. The actual simulation model has a much more extensive interface between the wake model and the AWC algorithm enabling a wide range of possible future applications, but is visualized here in a simplified way, sufficient for the present discussion. Finally, part of the AWC algorithm may operate at slower sampling rates (red lines). The main components are explained separately in more detail in the remainder of this section.

[Figure]

**Figure 1.** Block scheme of the simulation model

**2.1 Wind field**

[revised manuscript text omitted]

For the development and analysis of dynamic AWC algorithms, FarmFlow has been extended to model a quasi-dynamic flow in two-dimensional wind fields at hub height (see Sect. 2.1) with spatiotemporal variations. To this end, the wake generated by a wind turbine is propagated downstream  in such a way that it follows on its way the local wind direction variations  in the wind field. This way, both time delay and meandering effects are modelled. The simulation time is equal to the time step in the wind field time series.  Since the travel time of a wake between two turbines takes  longer than the  simulation sample time, the arriving wakes from upstream turbines need to be time synchronized with the departure of wakes in the current time window. Because the streamlines of the two-dimensional wind fields are curved and are varying in time, the trajectory of each wake needs to be corrected at the location of arrival, based on the trajectory of the undisturbed flow. After the correction, the wakes are stored in memory including the time and location of arrival. In summary, the quasi-dynamic wind farm simulation is realized as follows:

1.  For a given time instant, the simulation starts with the most upstream turbine with respect to the farm-average wind direction, and ends with the most downstream turbine, .

2.  For a given turbine,  all arriving wakes (if any), are determined using wake information calculated at previous time instances and stored in memory in Step 5

3. From the undisturbed wind field and arriving wakes,  a rotor averaged wind speed and wind direction is calculated.

4. Given the determined rotor averaged wind  conditions, and the yaw position (coming from the yaw system model  in Step 8, see also Fig. 1), the yaw misalignment angle is computed. Subsequently, the power   production and rotor thrust force are determined, and a wake calculation is started using the (stationary) wake model in FarmFlow  for the given turbine.

5. The wake is calculated until  reaches a downstream turbine  (if any). The path of the calculated wake is determined using the time varying wind direction

in the undisturbed wind field, resulting in wake meandering. The wake information is then stored in memory, including the time of arrival at the downstream turbine for use in the wake simulation of that turbine  at a later time instant in Step 2.

240   6. The simulation continues with the  following most upstream turbine until all turbines  have been simulated for the current time instant.

7. The input and output data for all turbines are written in an output file, which forms the interface to a dynamic link library (DLL) that implements the yaw system dynamics and the dynamic robust AWC algorithm.

8. The DLL is called, which updates the yaw  positions of all turbines, and communicates this
245   information back to FarmFlow. The simulation of the next time instant is initiated in Step 1.

[revised manuscript text omitted]

The dynamic robust AWC consists of the following blocks (see Fig. 1):

– LP filter: a second order Butterworth lowpass filter, the cutoff frequency of which will be varied to study its impact on the power gain and yaw duty. As visualized in Fig. 1, the LP filter acts on the wind speed and wind direction signals.

305 – Hysteresis: adding hysteresis on the (filtered) wind direction signal, centred at wind directions where the yaw misalignment set-points change sign, can reduce the yaw duty as demonstrated in earlier by Kanev (2020). The hysteresis logic for a given turbine is defined as

$$\mathcal{H}(b): \phi_{hyst}(k) = \begin{cases} \phi_{hyst}(k-1) & |\phi_{LP}(k) - \phi_c| \leq b \\ \phi_{LP}(k) & \text{otherwise} \end{cases}$$

wherein $\phi_c$ is any wind direction for which the yaw misalignment set-point for that turbine changes sign, $\phi_{LP}(k)$ is the LP filtered wind direction (input to the hysteresis), and $b$ defines the hysteresis size. Based on the findings in Kanev (2020), a wider range of hysteresis sizes are considered in the present work ($b$ up to $4°$).

- Sampling: limits the update rate of the yaw misalignment set-points, i.e. the frequency at which they are communicated to the yaw controller implemented in the yaw system model.

- LUT containing the yaw misalignment set-points for all wind turbines in the wind farm as function of the wind speed and wind direction, obtained through stochastic program that accounts for uncertainty in a number of parameters (see Sect. 3 for more details). The optimization is performed using the conventional (stationary) FarmFlow model for a range of wind conditions to populate the LUT.

The robustness of the AWC is realized through the robust design of the LUT, while the last three components of the AWC controller in the list above represent the dynamic adaptation algorithm. The robust design through stochastic programming will be discussed in Sect. 3. The quantities to which robustness is to be achieved are selected through uncertainty quantification in Sect. 3.2. The selection of dynamic adaptation algorithm parameters (LP filter cut-off frequency, hysteresis size and sample time) to optimize the balance between power gain and yaw duty is topic of Sect. 4.

**3 Robust AWC optimization**

The conventional approach to (nominal) AWC design involves the synthesis of LUT containing yaw misalignment set-points for the wind turbines in the farm, optimized for a range of input conditions. These include at least the wind direction (Kanev et al., 2018), but sometimes also the wind speed or other atmospheric conditions such as the turbulence intensity (Bossanyi, 2018; Doekemeijer et al., 2021). In this work, the yaw misalignment set-points  in the LUT will essentially be a function of the wind direction  only. The wind speed signal entering the block "Robust AWC (LUT)" in Fig. 1 is meant to indicate that AWC is only active in a certain range of below rated wind conditions (4-12 ms$^{-1}$ used here).

This section considers the synthesis of robust AWC in which a number of parameters is treated as uncertain, continuing along the lines of previous work by Quick et al. (2020). Here, a more extensive list of uncertain parameters is treated, consisting of quantities related to model parameter uncertainty, measurement uncertainty and input variability, all treated in a unified stochastic framework using PDFs (Sect. 3.1). The impact of these uncertainties on the optimal yaw set-points is quantified through forward uncertainty propagation (Sect. 3.2). To this end, the PDFs are sampled and the underlying simulations are performed using the conventional (static) FarmFlow wake model.  This allows to evaluate how much the optimal yaw set-points change as result of variations in the different uncertain parameters within their assumed uncertainty sets,

**Table 2.** Uncertain parameters considered in the uncertainty quantification framework, their assumed ranges and stochastic modelling

| Parameter | Uncertainty type | Range | PDF |
|---|---|---|---|
| Wind direction | Variability and measurement uncertainty | $[-10, 10]^\circ$ | Normal ($\mu = \phi_{LUT}, \sigma = 4.25^\circ$) |
| Wind speed | Variability | $[4, 12]$  $\text{ms}^{-1}$ | Weibull ($k = 2.24, A = 9.3$) |
| Turbulence intensity | Variability | $[4, 20]$ % | Weibull ($k = 3, A = 0.073$) |
| Yaw error | Variability and measurement uncertainty | $[-10, 10]^\circ$ | Laplace ($\mu = 0, \nu = 5^\circ$) |
| Wind shear exponent | Variability and model uncertainty | $[0.04, 0.20]$ | Normal ($\mu = 0.12, \sigma = 0.05$) |
| Air density | Variability | $[1.18, 1.38]$  $\text{kgm}^{-3}$ | Normal ($\sigma = 0.015$) |
| Yawed power loss exp. | Model uncertainty | $[1.3, 2.3]$ | biased inverse Gaussian ($\mu = 0.52, \lambda = 8$) |
| Thrust curve variation | Model uncertainty | $[-10, 10]$ % | Normal ($\mu = C_{t,nom}, \sigma = 10/3$ %) |
| Power curve variation | Model uncertainty | $[-5, 5]$ % | Normal ($\mu = C_{p,nom}, \sigma = 5/3$ %) |

so that it can be decided which uncertain parameters should be included into the robust optimization. In Quick et al. (2020) a similar analysis is performed by focusing on the impact of variations in the sizes of the uncertainty sets instead. Finally, 
[revised manuscript text omitted]

To solve the underlying optimization problems, a tailor made algorithm has been used that requires a minimum number of function evaluations (farm simulations) to converge. The algorithm is similar to the conventional bisection method, but generalized to multivariate objective functions. By confining the optimization variable to lie within an initial n-dimensional box, the gradient of the objective function is evaluated at the centre point at each iteration and the box is reduced in size by keeping only that part that is oriented opposite to the gradient. While this algorithm has no theoretical guarantees to converge to an optimum solution for general nonlinear functions, has been successfully used for many years by the authors and works pretty well for the application at hand, its low calculation effort being its main advantage over alternative algorithms. This allows it to be used in combination with relatively complex wake models such as FarmFlow. To reduce computation time even more, the number of optimization variables is limited to the yaw set-points of the two most upstream turbines in each row of turbines oriented downstream. The yaw set-points for the remaining turbines in the row are linearly decreased between the second turbine and the last one, which has zero yaw misalignment set-point. No limitation has been applied to the yaw set-points in this section.

The yaw misalignment angles are optimized for one uncertainty parameter at a time, keeping all other parameters at their nominal values. More precisely, let $p_i$ be a given parameter from the list in Table 2, $\mathcal{U}_i$ be the corresponding uncertainty range, and $\mathcal{D}_i(p_i)$ – its PDF. For a given sample $p_i^{(r)} \in \mathcal{U}_i$ of the parameter $p_i$, and keeping the remaining parameters at their nominal values (i.e. $p_j = p_j^{(nom)} \in \mathcal{U}_j$, $j \neq i$), conventional (non-robust) AWC design solves the optimization problem of finding the vector of best yaw misalignment set-points $\gamma = [\gamma_1, \ldots, \gamma_N]$, $N$ being the number of turbines, with respect to the total power

production of the wind farm, i.e.

$$\gamma_{det}(p_i^{(r)}) = \arg \underset{\gamma}{\min\max} \sum_{t=1}^{N} P_t\left(\gamma | p_i = p_i^{(r)}, p_j = p_j^{(nom)}, j \neq i\right).$$

Notice that each individual turbine's power production, $P_t$, may depend on the yaw misalignments of other turbines through the wake effects. The optimal power gain for sample $p_i^{(r)}$ is then defined as

$$\delta P_{opt}(p_i^{(r)}) = \left(\sum_{t=1}^{N} P_t\left(\gamma_{det}(p_r) | p_i = p_i^{(r)}, p_j = p_j^{(nom)}, j \neq i\right)\right) \bigg/ \left(\sum_{t=1}^{N} P_t\left(\gamma = 0 | p_i = p_i^{(r)}, p_j = p_j^{(nom)}, j \neq i\right)\right).$$

Table 3 gives for each parameter $p_i$ its uncertainty set $\mathcal{U}_i$, the selected step size in the uncertainty sampling (resulting in samples $p_i^{(r)}$, $r = 1, 2, \ldots$ for which the yaw misalignments are optimized), as well as the nominal values of the parameters ($p_j^{(nom)}$). The resulting optimal yaw misalignment angles $\gamma_{det}(p_i^{(r)})$ are statistically summarized with the box plots in Fig. 5. In the figure, three box plots per parameter are depicted, corresponding to the considered three cases. Each box plot gives the minimum and the maximum value (upper and lower line segment), the first and third quartile (lower and upper sides of the boxes), and the median (line segment inside the box).

The following observations can be made from the figure

– Variations in the power coefficient and the air density have no impact on the optimal yaw misalignment set-points, which is expected as these parameters have little to no influence on the wake deficits behind the turbines

– Wind direction variability has by far the largest impact on the optimal yaw misalignment set-points which, of course, is due to their very pronounced influence on the wake locations with respect to downstream turbines. Clearly, this parameter is the most important one to consider in a robust AWC setting. This conclusion confirms earlier observations by Quick et al. (2020).

– Other quantities, variations of which lead to significant changes in the optimal yaw set-points, are the wind speed, yaw error, turbulence intensity, and wind shear. Notice that for some turbulence intensity and wind speed cases the minimum yaw misalignment values are equal to zero, which occur for uncertainty samples around the edges of their ranges of variation. For measurable quantities (such as the wind speed), such values can better be excluded from the robust optimization when possible and be used instead to deactivate AWC.

– Variations in the thrust curve and the yaw-induced power loss exponent have generally limited impact on the optimal yaw set-points, which suggests that they could be left out from the robust optimization.

Notice that, due to the fact that no limitations have been imposed, the yaw misalignment set-points are getting quite high in some cases, raising up to values of 40° and even higher. Such a high yaw misalignments are currently considered unrealistic in a real-life application. Usually, they are limited to around 30° in many research studies, or to even lower values in the first field tests with wake redirection (Fleming et al., 2017, 2019; Doekemeijer et al., 2021). It becomes clear from Fig. 5 that by imposing a limitation of ±30° one would significantly limit the variation in the yaw misalignment set-points. Nevertheless, the

**Table 3.** Sampling of the uncertainties for the purpose of uncertainty quantification

| Parameter | Range | Step size | Nominal value |
|---|---|---|---|
| Wind direction | [-10, 10]° | 1° | 0°(Cases 1,2), 4°(Case3) |
| Wind speed | [4, 12] ms$^{-1}$ | 1 ms$^{-1}$ | 8 ms$^{-1}$ |
| Turbulence intensity | [4, 20] % | 1 % | 7 % |
| Yaw error | [-10, 10]° | 1° | 0° |
| Wind shear exponent | [0.04, 0.20] | 0.02 | 0.09 |
| Air density | [1.18, 1.38] kgm$^{-3}$ | 0.02 kgm$^{-3}$ | 1.225 kgm$^{-3}$ |
| Yawed power loss exp. | [1.3, 2.3] | 0.1 | 1.43 |
| Thrust curve variation | [-10, 10] % | 2 % | 0 |
| Power curve variation | [-5, 5] % | 1 % | 0 |

[Figure]

**Figure 5.** Box plots for the optimal yaw misalignment angles of the most upstream turbine, evaluated for the considered parameters and setup cases

main conclusions drawn above in terms of the most significant uncertainty contributors will still hold, with probably only the

460   wind shear disappearing from this list.

[revised manuscript text omitted]

– LP filter time constant: 60 s.

[Figure]

**Figure 10.** KPIs for the simulations with dynamic robust AWC: average and worst-case yaw travel, average and worst-case number of yaw start/stop events, and power gain per unity yaw travel and per unity yaw start/stop events

    – sample time: 10 s.

With these adaptation parameters, the following results are achieved by the optimized dynamic robust AWC algorithm in terms of energy gain and yaw duty for the considered case study, all expressed relatively with respect to reference case (AWC-free):

    – energy gain: relative increase of 12% in the wind farm energy production

    – average yaw travel: relative increase of just 1.13 (i.e. 13% increase), which is seems quite acceptable given the fact the AWC requires the nacelle to travel substantially between positive and negative offsets as the wind direction changes.

- worst-case yaw travel: relative change of 0.95 (i.e. 5% reduction) in worst-case yaw travel. Since in the reference case it is the last turbine in the row that gets worst-case yaw travel in the simulation, this result shows that it remains higher than the yaw travel of the first four turbines even under dynamic robust AWC.

- average number of yaw events: relative change of 0.51 (i.e. 49% reduction) in average number of start/stop yaw actions. Having significantly less start/stop events  in the reference case  might first seem counter-intuitive, but does happen. The reason for this is the negative slope in the yaw misalignment setpoints to the right of their maximum value (and to the left of their minimum value), see Fig. 11, which has a damping effect on the yaw motion. It can be observed, for instance, in the left-hand side plot Fig. 8 for the first turbine in the row (compare the solid black line with the dashed black line). For the remaining turbines (not plotted here) this effect is even more pronounced as they are yawed more often in the reference case.

- worst-case number of yaw events: relative change of 0.67 (i.e. 33% reduction) in worst-case number of yaw events.

Altogether, it can be concluded that the results are rather positive with dynamic robust AWC achieving high energy gain and overall reduction in the number of yaw events, at quite limited negative impact on the average yaw travel.

**5 Case studies**

This section presents the results from two case studies. The first one represents a example of robust AWC design performed for a simple farm consisting of a few turbines in a row. The second one represents a realistic case study with dynamic robust AWC applied to an offshore wind farm.

**5.1 Simplified example of robust AWC design**

To exemplify the robust AWC design, consider the single-row layout consisting of five 3 MW wind turbines, with 90 m rotor diameter, separated at a distance of 7D (equivalent to the setup in Cases 2 and 3 in Sect. 3.2). The robust optimization problem in Eq. (3) is solved for wind directions ranging from 248° to 292° at a step of 1°, an interval centred around the row orientation of 270°. In addition to that, a nominal AWC optimization is performed for the nominal values of the uncertainties $p^{(nom)} \in \mathcal{U}$,

$$\gamma_{nom}(\phi_{LUT}) = \arg\max_{\gamma} \sum_{t=1}^{N} P_t\left(\gamma | p^{(nom)}, \phi_{LUT}\right), \tag{4}$$

[revised manuscript text omitted]